# On the Joint Interaction of Models, Data, and Features

**Yiding Jiang**
Carnegie Mellon University
yidingji@cs.cmu.edu

**Christina Baek**
Carnegie Mellon University
kbaek@cs.cmu.edu

**J. Zico Kolter**
Carnegie Mellon University
Bosch Center for AI
zkolter@cs.cmu.edu

## Abstract

Learning features from data is one of the defining characteristics of deep learning, but the theoretical understanding of the role features play in deep learning is still in early development. To address this gap, we introduce a new tool, the *interaction tensor*, for empirically analyzing the interaction between data and model through features. With the interaction tensor, we make several key observations about how features are distributed in data and how models with different random seeds learn different features. Based on these observations, we propose a conceptual framework for feature learning. Under this framework, the expected accuracy for a single hypothesis and agreement for a pair of hypotheses can both be derived in closed form. We demonstrate that the proposed framework can explain empirically observed phenomena, including the recently discovered Generalization Disagreement Equality (GDE) that allows for estimating the generalization error with only unlabeled data. Further, our theory also provides explicit construction of natural data distributions that break the GDE. Thus, we believe this work provides valuable new insight into our understanding of feature learning.

## 1 Introduction

It is commonly said that deep learning performs feature learning, whereby the models extract useful patterns from the data and use the patterns to make predictions. Most successful applications of deep learning today involve first training the models on a large amount of data and then fine-tuning the pre-trained model on downstream tasks (Chen et al., 2020; Brown et al., 2020; Radford et al., 2021). Their success suggests the models are learning useful and transferable knowledge from the data that allows them to solve similar tasks more efficiently. Experimentally, many different works (Nguyen et al., 2016; Zeiler and Fergus, 2014; Bau et al., 2017; Olah et al., 2017; 2018; Li et al., 2015) have studied various aspects of the features learned by deep neural networks. These works help the community gain a better intuitive understanding of the mechanisms underpinning deep learning as well as improve the interpretability of deep models. However, to the best of our knowledge, the theoretical understanding of the role features play in deep learning is still under-explored. For example, the popular neural tangent kernel (NTK) (Jacot et al., 2018) framework analyzes deep learning as kernel regression with features defined by the model's gradient *at initialization*, which is independent of the training data.

While it may be intuitive to think of defining features as quantifying the "*unit*" of information in data that models use to make predictions, the community has yet to reach a consensus on the exact definition of features in deep learning beyond toy models. Nonetheless, it is undeniable that the models have learned *something* from the data. In fact, the same models trained with different random seeds would learn different information that leads to different predictions (Lakshminarayanan et al., 2017). This phenomenon has important downstream consequences for ensembling randomly initialized networks including better generalization (Allen-Zhu and Li, 2020), calibration (Lakshminarayanan et al., 2017), and the Generalization Disagreement Equality (GDE) (Jiang et al., 2022) where the expected test accuracy is equal to the expected agreement in deep ensembles.

We postulate that a good definition of features should be fine-grained enough to differentiate the difference in the knowledge of different models. To this end, we attempt to define such a construct that allows us to analyze the similarities and differences of information learned by different models, while also remaining amenable to quantitative and theoretical analysis. Through this definition, we

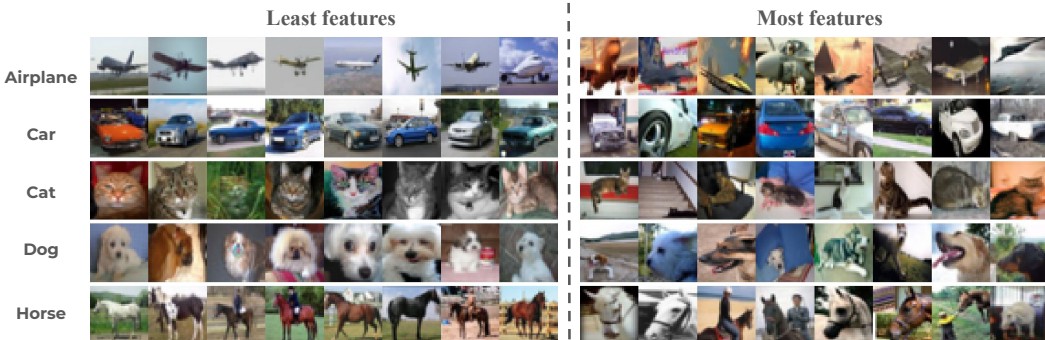

Figure 1: Visualization of images with *the least features* (left) and *the most features* (right) for classes of CIFAR 10 under our feature definition (defined in Section 3.1). Each row corresponds to one class of CIFAR 10 (zoom in for better viewing quality). We can see that the images with the least features are semantically similar to each other whereas the images with the most features are much more diverse and contain unusual instances of the class or objects in rare viewpoints. More examples for all classes can be found in Figure 12 of the appendix.

can gain insights into the behavior of models and the underlying mechanisms of feature learning. Notably, we show that GDE arises immediately as a consequence of how neural networks learn appropriately defined features. This phenomenon was previously explained by *assuming* calibration of the underlying ensemble, which is often a strong assumption to make.

We begin with an empirical investigation of feature learning, using a natural definition of features on real data (Figure 1) that allows us to easily compare information about the data distribution learned by different models and a construction we propose called the *interaction tensor*. We define features to be the projection onto the principal components of different models' last-layer activation. The interaction tensor then jointly models the features learned by multiple models and across multiple data points. Inspecting this tensor constructed on collections of models, we find that the model of data presented in Allen-Zhu and Li (2020) is not conceptually reflective of the actual observed phenomena, suggesting that an alternative model is needed.

Based on these observations, we propose an alternative (still simplified) model of feature learning, which better captures the above phenomenon. Specifically, we posit a framework where features come in two types: dominant (more frequent) and rare (less frequent), which captures the observed heavy-tailed nature of features. We also assume that *data points* either contain a small number of dominant features or a large number of rare features and that models learn features according to their frequency in the data set; this captures the observed phenomenon where data points with fewer features receive higher-confidence predictions. Under this model, we can analytically derive expressions for the accuracy and agreement of resulting classifiers. Despite the simplification, we show that our framework naturally captures the phenomenon of GDE without assuming calibration. Instead, we show that GDE arises immediately as a consequence of the distributional properties of features in natural data. Finally, we demonstrate that the framework can make accurate predictions about the effects of merging classes and changing data distribution on GDE and calibration, leading to the construction of natural data distributions that break GDE.

Thus, we believe overall that our framework of feature learning shows promise as an additional useful and valuable conceptual tool in understanding how deep learning works. Note that we do *not* attempt to derive how our model can arise mechanistically via optimization, but we believe that this can be considered a strength of our approach. Similar to natural sciences and econometrics, the empirical phenomena of deep learning can be phenomenologically understood "on their own terms" at many different layers of abstraction, and our model provides one such formalism that comports with observed behavior at the level of *observed* feature learning phenomena.

## 2 RELATED WORKS

**Feature learning.** Representation learning (Bengio et al., 2013) is the practice of discovering useful features from raw data directly instead of using hand-crafted features. Deep learning is the de facto approach for learning features from a large amount of data (Chen et al., 2020; Brown et al.,

2020; Radford et al., 2021), but a formal understanding of the process is still lacking. Recent works have started to incorporate feature learning into theoretical analysis (Li et al., 2019; Allen-Zhu and Li, 2020; Yang and Hu, 2021; Karp et al., 2021; Wen and Li, 2021; Allen-Zhu and Li, 2022; Ba et al., 2022). This paper is most immediately related to Allen-Zhu and Li (2020) which proposes the *multi-view* data structure where there exist two types of data: *multi-view* data which contain all the features of a class and *single-view* data which contain only one feature. In this work, we investigate whether this structure of features holds in practice by treating features as first-class citizens in both empirical investigation and theoretical analysis. Our experimental results reveal a more nuanced perspective on the structure of data and "features". Based on these observations, we propose an alternative conceptual model that better reflects how features, data, and models behave in reality. We analyze the generalization property of the model and also its *agreement property* (Nakkiran and Bansal, 2020). We also show that this feature learning model provides an alternative condition under which the curious GDE phenomena observed in Jiang et al. (2022) can arise.

**Ensemble and Generalization Disagreement Equality.** Deep ensembles (Lakshminarayanan et al., 2017), trained with different initializations on the same dataset, have demonstrated superior performance over classical methods such as bagging and Bayesian approaches (Breiman, 1996; Welling and Teh, 2011; Gal and Ghahramani, 2016). One of the notable attributes of deep ensembles is their well-calibrated nature — making predictions with appropriate confidence levels (Murphy and Epstein, 1967). Jiang et al. (2022) explored the implications of this calibration by showing that test accuracy can be estimated using the agreement of two independent models (Nakkiran and Bansal, 2020), a phenomenon that the authors called the Generalization Disagreement Equality (GDE). However, calibration is a strong assumption – why deep ensembles are calibrated is largely an open problem (Fort et al., 2019). To address this gap, this work shows that the phenomenon of GDE can arise within a feature learning context, without making explicit calibration assumptions.

**Understanding representations.** One line of work tries to understand deep learning with a more empirical approach. Many try to understand the features by visualizing what aspects of the input data they correspond to (Nguyen et al., 2016; Zeiler and Fergus, 2014; Bau et al., 2017; Olah et al., 2017; 2018). Another line of work attempts to compare representations of different models (Li et al., 2015; Raghu et al., 2017; Morcos et al., 2018; Kornblith et al., 2019). We demonstrate that simple PCA can reduce the redundancies in high-dimensional representation and find a parsimonious set of features that the model relies on to make predictions. Our feature clustering algorithm may be seen as a generalization of pair-wise matching proposed by Li et al. (2015).

## 3 CONNECTING MODELS AND DATA WITH FEATURES

In this section, we describe the procedure for constructing the interaction tensor $\boldsymbol{\Omega} \in \{0, 1\}^{M \times N \times T}$. The first axis corresponds to $M$ models, the second axis corresponds to $N$ data, and the last axis corresponds to $T$ features. If the $n^{\text{th}}$ data point contains the $t^{\text{th}}$ feature and the $m^{\text{th}}$ model has learned the $T^{\text{th}}$ feature, then $\boldsymbol{\Omega}_{mnt}$ would be 1. This tensor captures how features are distributed (see Figure 6 for an illustration). First, we identify features within each model of ensembles. Then, we cluster the features and construct the interaction tensor with the identified feature clusters.

**Notations.** Let $x$ denote a point in $\mathcal{X}$, the input space, and $y \in [C]$ denote the label, where $[C]$ is the set of labels, $[1, 2, \ldots, C]$. Let $\mathscr{D}$ be the data distribution over $\mathcal{X} \times [C]$. We use $(x, y)$ to denote samples from the random variable following $\mathscr{D}$. Let $f : \mathcal{X} \to [C]$ be a parameterized function, $f(x) = (\psi \circ \varphi)(x)$. $\varphi : \mathcal{X} \to \mathbb{R}^d$ maps the input $x$ to a $d$-dimensional representation, and $\psi : \mathbb{R}^d \to [C]$ maps the representation to a class. For a collection of models $\{f_1, \ldots, f_M\}$, we use *confidence* to denote their average one-hot predictions for the ground truth label.

### 3.1 PRINCIPAL COMPONENTS OF ACTIVATION AS FEATURES

Many works (Li et al., 2015; Olah et al., 2017) study the representations of deep neural networks by treating individual neurons as features or the most elementary unit of the representation, but these representations are high-dimensional vectors, which contain redundant information. Furthermore, a single feature may be distributed across multiple neurons. Instead, ideal features should be parsimonious and can capture the dependencies between different coordinates of the representations. These criteria can be fulfilled by dimensionality reduction methods. We choose *principal component analysis* (PCA)

which captures the linear dependencies between different coordinates of the representations and can be efficiently computed with stochastic algorithms (similar to Härkönen et al. (2020)). While this procedure can be applied to any layer, we use the last layer representation to avoid non-linear interaction through superposition (Elhage et al., 2022). Concretely, given a neural network and a set of data points $\mathbf{X} = \left[x^{(1)}, x^{(2)}, \ldots, x^{(N)}\right]^\top$, we use $\Phi \in \mathbb{R}^{N \times d}$ be the matrix that contains all of $\varphi\left(x^{(i)}\right)$ as its rows. The singular value decomposition (SVD) yields $\Phi = \mathbf{U}\mathbf{\Sigma}\mathbf{V}^\top$ where the columns of $\mathbf{V} \in \mathbb{R}^{d \times d}$ contain the principal components of $\Phi$. We use the top $K$ principal components $\mathbf{V}_{:K}$ and project the representations to $\mathbb{R}^K$, $\Phi^{\text{proj}} \triangleq \Phi\mathbf{V}_{:K} = \left[\mathbf{V}_{:K}^\top\varphi\left(x^{(1)}\right), \mathbf{V}_{:K}^\top\varphi\left(x^{(2)}\right), \ldots, \mathbf{V}_{:K}^\top\varphi\left(x^{(N)}\right)\right]^\top$. For notation simplicity, we will use $\upsilon(x)$ to denote $\mathbf{V}_{:K}^\top\varphi(x)$ and $\upsilon_{m,k}(x)$ to denote the $k^{\text{th}}$ entry of of the $m^{\text{th}}$ model's $\upsilon(x)$. Intuitively, we can interpret the principal components as a feature, or more concretely, orthogonal subspaces that the model uses to classify any given data points in $\mathbf{X}$.

## 3.2 CONSTRUCTING THE INTERACTION TENSOR

**Clustering features of different models.**   Given $M$ models, $\{f_1, f_2, \ldots, f_M\}$, we can compute the projected representation for each network of the $M$ models, $\left\{\Phi_1^{\text{proj}}, \Phi_2^{\text{proj}}, \ldots, \Phi_M^{\text{proj}}\right\}$. For a single model $f_m$ and its $k^{\text{th}}$ feature, we can compute its mean and variance:

$$\mu_{m,k} \triangleq \mathbb{E}_{(x,y)\sim\mathscr{D}}\left[\upsilon_{m,k}(x)\right], \ \sigma_{m,k}^2 \triangleq \mathbb{E}_{(x,y)\sim\mathscr{D}}\left[\left(\upsilon_{m,k}(x) - \mu_{m,k}\right)^2\right]. \tag{1}$$

For models $(f_i, f_j)$ and their respective $a^{\text{th}}$ and $b^{\text{th}}$ features, we can define their correlation to be:

$$\rho_{(i,j),(a,b)} \triangleq \mathbb{E}_{(x,y)\sim\mathscr{D}}\left[\left(\upsilon_{i,a}(x) - \mu_{i,a}\right)\left(\upsilon_{j,b}(x) - \mu_{j,b}\right)\right]\left(\sigma_{i,a}\ \sigma_{j,b}\right)^{-1}. \tag{2}$$

This can be seen as performing the procedure of Li et al. (2015) with PCA projected representations. We use $\mathbf{K}_{i,j} \in [-1,1]^{K \times K}$ to denote the collection of all pair-wise correlation values between the features of $f_i$ and $f_j$, and $\mathbf{\Lambda} \in [-1,1]^{M \times M \times K \times K}$ to denote the collection of all the correlation matrices between every pair of models. A centralized list of notations is provided in Appendix A.

With $\mathbf{\Lambda}$, we can identify unique *feature clusters* in the $MK$ features learned by all models. To account for the arbitrary direction of correlation in $\mathbf{\Lambda}$, we take the absolute value and use a threshold, $\gamma_{\text{corr}} \in (0,1)$, to determine whether two features should be considered as the same feature. We use a greedy clustering algorithm (Algorithm 1) to match the features with one another, as k-partite matching is known to be NP-complete for k > 2 (Garey and Johnson, 1979). After running the clustering algorithm, each feature is assigned to one of $T$ clusters (where $T \le MK$), and we treat every feature in a single cluster as the same feature. The greedy algorithm is efficient and does not generate a fixed number of clusters, which is desirable in cases where some features have low correlations with other features and should be isolated as a unique cluster. More sophisticated algorithms, such as graph cut, could be used, but we find the greedy algorithm sufficient for our purposes. See Algorithm 1 and Appendix E.2 for details on the algorithm and hyperparameters.

**Matching features to data points**   Once the features of all models are clustered, we can identify which features are present in each data point of $\mathbf{X}$. First, we normalize each individual $\upsilon_{m,k}$ by its $\ell_\infty$-norm, which ensures that all features are between 0 and 1. We will denote the row-normalized $\Phi_m^{\text{proj}}$ as $\widehat{\Phi}_m^{\text{proj}}$. We then pick another threshold, $\gamma_{\text{data}} \in (0,1)$, that decides whether a feature is present in a data point. Concretely, if the $k^{\text{th}}$ entry of the $n^{\text{th}}$ row in $\widehat{\Phi}_m^{\text{proj}}$ is larger than $\gamma_{\text{data}}$, we assign to the $n^{\text{th}}$ data point in $\mathbf{X}$ the feature cluster containing the $m^{\text{th}}$ model's $k^{\text{th}}$ feature. In other words, if the $m^{\text{th}}$ model's $k^{\text{th}}$ feature belongs to the $t^{\text{th}}$ cluster, we say the $n^{\text{th}}$ data point contains the $t^{\text{th}}$ feature. In Figure 1, we visualize the data points with the most and least number of features.

**Aggregating Information.**   After thresholding, we have enough information to construct the interaction tensor. Each entry indicates whether the $t^{\text{th}}$ feature is present in *both* the $m^{\text{th}}$ model and $n^{\text{th}}$ data point. In the next section, we will inspect various aspects of the interaction tensor $\mathbf{\Omega}$ and other experimental artifacts to understand how the models learn features from the data.

## 4 EXPERIMENTS AND OBSERVATIONS

**Experimental setup.**   We use a collection of $M = 20$ ResNet18 (He et al., 2016a) trained on the CIFAR-10 dataset (Krizhevsky et al., 2009) following the experimental set up of Jiang et al.

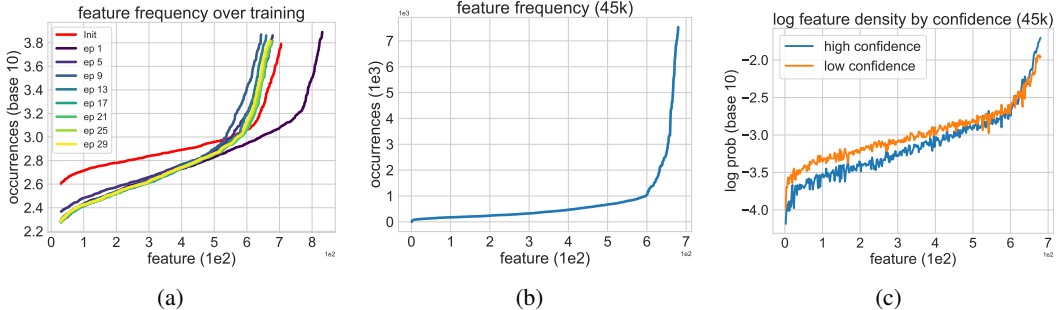

Figure 2: **(a)** Feature frequency over the course of training. The red curve represents the feature frequency at the initialization. **(b)** Features vs how often they appear in the dataset. The features are sorted by frequency and the distribution appears to be long-tailed. **(c)** Feature frequency by different confidence levels. Low-confidence data tend to have more low-frequency features.

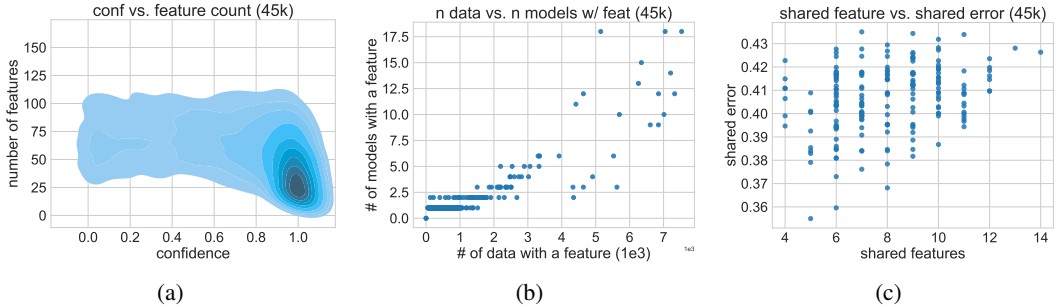

Figure 3: **(a)** Density estimation of confidence vs number of features over data. High-confidence data tend to have fewer features. **(b)** Scatter plot of the number of data with a feature vs. the number of models with that feature. The strong positive correlation suggests that the more data has a certain feature, the more likely the model will learn that feature. **(c)** Number of shared features vs shared error. The lower bound of shared error monotonically increases with the number of shared features.

(2022). The models are trained on random 45000 subsets of the whole training set (`45k`). In addition, we repeat the experiments on CIFAR-10 using random 10000 subsets (`10k`) and the SVHN dataset (Netzer et al., 2011) using random 45000 subsets of the whole training set (`SVHN`). We compute $\mathbf{\Lambda}$ and $\mathbf{\Omega}$ on the test set using the output of the penultimate layer (i.e., $\psi$ is the final linear layer). The experimental details can be found in Appendix E. We primarily show `45k` here and leave the `10k` and `SVHN` results to Appendix F, both of which are similar to the `45k` results qualitatively.

**Observation 1 : Feature frequency is long-tailed.** **(O.1)** We use the interaction tensor $\mathbf{\Omega}$ to compute the frequency of each of the $T$ features in the dataset. First, we sum $\mathbf{\Omega}$ over the model axis and then clip the values to $1$ (since we are only interested in the relation between data and features here). Then, we sum over the data axis to obtain the number of data points that contain each individual feature. The features are sorted by their frequencies, which reveals a long-tailed distribution (Figure 2b). Furthermore, this distribution is a consequence of learning. In Figure 2a, we compare the feature frequency computed from 20 models over the first 30 epochs of training (i.e., 600 checkpoints) with the feature distribution for untrained models. We observe that untrained models have much higher frequencies for tail features (low frequency). After even a single epoch, the frequency of tail features decreases significantly and then stays relatively stable over training. At the head of the distribution, the models first learn a large number of features and then prune the features as training continues, eventually converging to a fixed distribution with a small number of features.

**Observation 2: The ensemble tends to be more confident on data points with fewer features, and data points with lower confidence tend to have more features with low density.** **(O.2)** Another question that we want to understand is how the features interact with the confidence of the ensemble. In Figure 3a, we show the joint density plot of the ensemble's confidence for a data point and the number of features the data point has for `45k`. We can see that low-confidence data tend to have more features, whereas high-confidence data points tend to have fewer features. This finding

contradicts the model of Allen-Zhu and Li (2020) in which if a data point is multi-view (i.e., contains more features), all members of the ensembles will classify it correctly (i.e., high confidence). A plausible explanation for this observation is that there is a small sub-population of features that are learned by a large number of models. Based on O.1, we postulate that these features are learned by more models because they appear with higher probability. Furthermore, we plot the log density of features[1] in high-confidence data and all other data points (low-confidence) in Figure 2c. We see that the low-confidence data tend to have more features with low density in Figure 2b. One explanation is that the features in the tail are responsible for different models making different predictions.

**Observation 3: Number of models with a certain feature is positively correlated with the feature's frequency.** (**O.3**) The interaction tensor also reveals how the number of data points containing a certain feature relates to the number of models that have learned that feature. This relationship can shed light on how models learn features of different frequencies in practice. In Figure 3b, we observe that the number of models with any given feature has a strong positive correlation with the frequency of that feature appearing in the data (linear / super-linear). This implies that the more a feature appears in the data, the more likely a model will pick it up. We hypothesize that the feature learning procedure can be phenomenologically approximated by a sampling process where the probability of learning a feature is related to how often that feature appears in the data.

**Observation 4: Models with similar features make similar mistakes.** (**O.4**) Another natural hypothesis is that models with similar features should make similar mistakes. For every pair of models, we compare how many features they share to how often they make the same mistake relative to the average number of mistakes both models make (Figure 3c). We can see that the lower bound of shared error is almost *monotonically* increasing with the value of shared features. On the other hand, when models share a smaller number of features, the shared errors have a much larger variance, which indicates their errors are less correlated and therefore more random. This effect is more amplified for *different architectures*. We show this result on more than 20 diverse architectures in Appendix F.6.

Finally, we provide more analysis on the effect of using PCA for clustering in Appendix F.2 and different hyperparameters for the interaction tensor in Appendix F.10. We also explore the properties of features found under our definition in Figure 1 and Appendix F.3. In Appendix F.11, we investigate the feature distributions at different depths and discuss the interpretation of these features.

## 5 A COMBINATORIAL FRAMEWORK OF FEATURE LEARNING

In this section, we present a new framework of feature learning for a binary classification based on the insights from the experiments. We saw in O.1 that the distribution of features is long-tailed. This means a relatively small number of unique features constitute a large proportion of all the features in the data. To facilitate analysis, we will assume there are two types of features, *dominant features* and *rare features*, where the dominant features appear with much higher probability than rare features. Further, we observed that data with high confidence tend to have much fewer features than the ones with high confidence (O.2). To model this behavior, we will assume that there are two types of data points: *dominant data* and *rare data*. The former contains a small number of dominant features and the latter contains a larger number of rare features. This is another simplification based on O.2, which shows that high-confidence data tend to have fewer high-frequency features.

**Definitions and additional notations.** Before describing the full model, we first define the parameters of the model as well as some additional notations: $\mathbf{p_d}$: the proportion of all data that are dominant, $\mathbf{p_r}$: the proportion of all data that are rare (this parameter is equal to $1 - p_d$), $\mathbf{c}$: the total model capacity which represents how many features a single model can learn, $\mathbf{t_d}$: the total number of dominant features available in the data for one class, $\mathbf{t_r}$: the total number of rare features available in the data for one class, $\mathbf{n_d}$: the total number of dominant features a single dominant data point has ($n_d \leq t_d$), and $\mathbf{n_r}$: the total number of rare features a single rare data point has ($n_r \leq t_r$). We will use $\bar{\Psi}(\cdot)$ to denote the set of all features a model or a feature has (an expanded list in Appendix A).

**Data generating process.** We can see the data generating process as the following sampling procedure. First, we decide which class the data point belongs to. We consider a class-balanced

---

[1]The features are plotted in the same order as Figure 2c (hence the jaggedness), and the density is obtained by normalizing with the total number of features in both high confidence and low confidence groups of data.

binary classification problem so each class occurs with an equal probability of $\frac{1}{2}$. Then, we decide whether the data point is dominant or rare. This is equivalent to sampling from a Bernoulli distribution, $\text{Ber}(p_d)$. If the data point is dominant, we sample $n_d$ dominant features uniformly *without replacement*. Vice versa, if the data point is rare, we sample $n_r$ dominant features uniformly *without replacement*. It is easy to verify that the proportion of dominant data points and features is $p_d$ and the proportion of rare data points and features is $p_r$.

**How the models learn.** We saw in O.3 that the frequency of features occurring in different models is positively correlated with the frequency at which the features occur in the data. We can model the learning process as another *sampling-without-replacement* process where the probability that a model learns a feature is *proportional* to the frequency at which the feature occurs in the data. Under this assumption, in expectation, $c_d = \frac{1}{2}p_d c$ of the features in a single model would be dominant features for a single class, and $c_r = \frac{1}{2}p_r c$ of the features for a single class would be rare. We can further simplify this process by assuming that the model will always sample $c_d$ dominant features for each class and $c_r$ rare features for each class[2].

**How the models make predictions.** For a data point $x$ and a model $f$, we assume that the model will correctly classify $x$ if the overlap between the features of $x$ and the features of $f$ is not empty (similar to assumptions in Allen-Zhu and Li (2020)). Otherwise, the model will perform a random guess. The expected error that a single model $f$ makes on a single datum pair $(x, y)$ is thus $\text{err}(f, x, y) = \frac{1}{2}\mathbb{1}\{\Psi(f) \cap \Psi(x) = \varnothing\}$. Further, given a pair of models $(f, g)$ and a single datum pair $(x, y)$, there are three distinct behaviors for how they will make predictions. **(1)** The two models will always agree with each other if both of them share feature with $x$, since both will classify $x$ correctly (i.e., if $|\Psi(f) \cap \Psi(x)| > 0$ and $|\Psi(g) \cap \Psi(x)| > 0$). **(2)** If the models both do not share any features with $x$, then by the previous assumptions, the models will make random guesses (see Appendix D.2 for why this is justified); however, if the models share features with each other, their random guesses will not be independent of each other (O.4). We hypothesize that how two models agree with each other is a function of $k$, the number of features they share, and $c$, the model capacity. We capture this intuition with an *agreement function*, $\zeta : \mathbb{N} \times \mathbb{N} \to [0, 1]$, which returns the probability that two models will agree based on how many features they share relative to the full model capacity. This function is crucial for understanding how models make mistakes. **(3)** Finally, if the models do not share any features with each other or with $x$, both models will perform independent random guesses, in which case they will agree $50\%$ of the time.

It is natural to ask how reasonable the simplifications are. In Appendix D, we discuss these simplifications (e.g., random guess and number of classes) in detail and provide a comparison between this framework and Allen-Zhu and Li (2020). We encourage interested readers to read this section. Still, we will see that this relatively simplified model readily offers interesting insights into observed phenomena and can make surprisingly accurate predictions about the results of experiments a priori.

## 5.1 ANALYTICAL FORMS OF ACCURACY AND AGREEMENT

Using this model, the *closed-form* form of expected accuracy, Acc, and expected agreement rates, Agr, can be derived through combinatorics. All propositions are proven in Appendix C.

**Proposition 5.1.** *The expected accuracy over the model distribution and data distribution is:*

$$\text{Acc} = p_d \left(1 - \frac{1}{2}\frac{\binom{t_d - c_d}{n_d}}{\binom{t_d}{n_d}}\right) + p_r \left(1 - \frac{1}{2}\frac{\binom{t_r - c_r}{n_r}}{\binom{t_r}{n_r}}\right). \tag{3}$$

**Proposition 5.2.** *Let $\binom{n}{r} = 0$ when $n < 0$, $r < 0$ or $n < r$, and let:*

$$q_1 = p_d \left(1 - \frac{\binom{t_d - c_d}{n_d}}{\binom{t_d}{n_d}}\right)^2 + p_r \left(1 - \frac{\binom{t_r - c_r}{n_r}}{\binom{t_r}{n_r}}\right)^2,$$

$$q_2(k) = p_d \frac{\binom{t_d - n_d}{c_d}^2}{\binom{t_d}{c_d}^2}\left(\sum_{a+b=k} \frac{\binom{c_d}{a}\binom{t_d - n_d - c_d}{c_d - a}}{\binom{t_d - n_d}{c_d}}\frac{\binom{c_r}{b}\binom{t_r - c_r}{c_r - b}}{\binom{t_r}{c_r}}\right) + p_r \frac{\binom{t_r - n_r}{c_r}^2}{\binom{t_r}{c_r}^2}\left(\sum_{a+b=k} \frac{\binom{c_d}{a}\binom{t_d - c_d}{c_d - a}}{\binom{t_d}{c_d}}\frac{\binom{c_r}{b}\binom{t_r - n_r - c_r}{c_r - b}}{\binom{t_r - n_r}{c_r}}\right),$$

*then the expected agreement between an i.i.d pair $(f, g)$ drawn for the model distribution is:*

$$\text{Agr} = \frac{1}{2} + \frac{1}{2}q_1 + \sum_{k=1}^{c}\left(\zeta(k, c) - \frac{1}{2}\right)q_2(k). \tag{4}$$

---

[2]Both $c_r$ and $c_d$ are rounded to the nearest integer such that the total number of features in a model is still $c$.

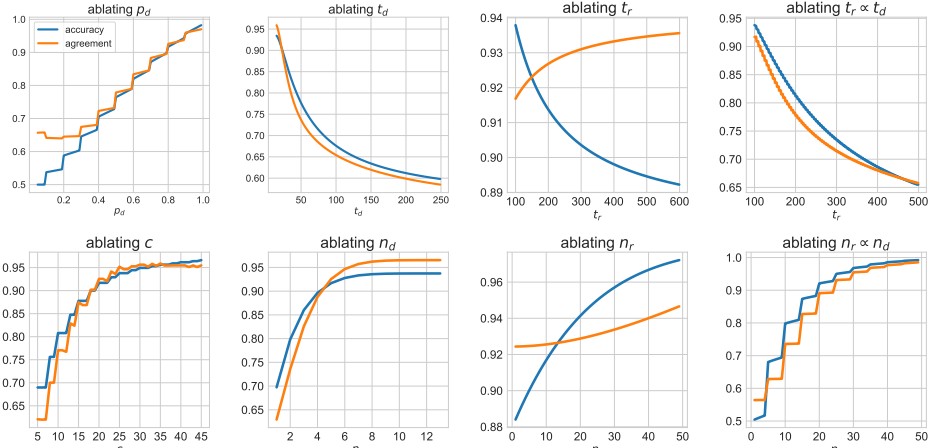

Figure 4: Numerical simulations using the analytical forms of accuracy and agreement. Each plot in the left three columns ablates a single parameter in the framework. The right column shows the effect of coupling two parameters, namely $(t_r, t_d)$ and $(n_r, n_d)$. Changing $t_r$ and $n_r$ alone deviates from GDE, but if they are coupled with dominant features, we can recover GDE (approximately).

Further, we discuss irreducible error and other properties of this framework in Appendix B and discuss a potential connection between feature learning and data scaling (Hestness et al., 2017).

## 5.2 NUMERICAL SIMULATION

We now study the properties of the analytical forms of the expected agreement and the expected accuracy. Instead of bounding their difference, we will use numerical simulation to characterize their properties and differences. The model has 6 free parameters, namely, $p_d, c, t_d, t_r, n_d, n_r$. We first pick a set of initial values and then vary each value to study the behavior of the model. Unless specified otherwise, the initial values used for all simulations are $p_d = 0.7, c = 20, t_d = 20, t_r = 180, n_d = 5, n_r = 10$. Further, we pick $\zeta(k, c) = 0.9\,\mathbb{1}\{k > 0\}$. This reflects the intuition that if two models share any features, then with high probability they would agree with each other. This is reasonable if we assume all the models in the same hypothesis distribution are naturally more inclined to agree with each other (O.4 and Figure 3c show that for a *10-class* classification the shared error is at least 35%). We study the properties and effects of $\zeta$ in Appendix F.7.

In the left 3 columns of Figure 4, we vary each parameter of the framework over a wide range of values. We observe that for both $p_d$ and $c$, the agreement closely tracks each other for a large portion of the parameter values. This suggests that the difference between generalization error and agreement is robust to how much of the data has dominant features and the size of the model. Further, we observe both accuracy and agreement saturate as the model capacity, $c$, increases. This is equivalent to increasing the model capacity with an infinite amount of training data. This is consistent with prior works on *model scaling* (Tan and Le, 2019) which suggests that model size may be related to how many features the model can learn. On the other hand, the behaviors of agreement and accuracy appear to be more sensitive to the other parameters that describe the relationship between total numbers of existing features and how often these features appear in a single data point, in particular $t_r$ and $n_r$, quantities that govern the distribution of rare data. Intuitively, the rare features and data represent the part of data distribution that appears in the tail of the data distribution and require *memorization* to learn (Feldman, 2020).

The observations about $n_r$ and $t_r$ suggest that GDE requires some distributional assumptions on the features and data in our framework (also in Jiang et al. (2022)). One possible hypothesis is that the relationship between $t_r$ and $t_d$ and the relationship between $n_r$ and $n_d$ follow the Pareto principle (Pareto, 1964) (given the long-tailed behavior in O.1). To verify the effect of this hypothesis, we vary $n_r$ and $t_r$ while keeping $n_d$ and $n_r$ proportional to them, that is, $n_d = \lfloor \alpha n_r \rfloor$ and $t_d = \lfloor \alpha t_r \rfloor$. We choose $\alpha = 0.2$ and show the results in the right column of Figure 4. Notice that if the ratio between these quantities is constant, agreement once again tracks the accuracy closely, indicating that the relationship between dominant and rare data is central to the origin of GDE in this model.

Table 1: Average accuracy and agreement on datasets with different interventions. $\widetilde{K}$ represents different number of superclasses and $\widetilde{K} = 10$ is the original CIFAR10. $G_i$ presents the data in the $(i-1) \times 20^{\text{th}}$ to $i \times 20^{\text{th}}$ percentile of blue intensity. The difference between accuracy and agreement is approximately the same for different $\widetilde{K}$'s (thus GDE holds) but not for different $G_i$'s.

| | $\widetilde{K} = 2$ | $\widetilde{K} = 3$ | $\widetilde{K} = 5$ | $\widetilde{K} = 10$ | $G_1$ | $G_2$ | $G_3$ | $G_4$ | $G_5$ |
|---|---|---|---|---|---|---|---|---|---|
| Accuracy | 0.80 | 0.84 | 0.78 | 0.81 | 0.62 | 0.60 | 0.59 | 0.66 | 0.70 |
| Agreement | 0.85 | 0.88 | 0.83 | 0.85 | 0.70 | 0.67 | 0.69 | 0.71 | 0.75 |
| Difference | 0.05 | 0.04 | 0.04 | 0.05 | 0.08 | 0.07 | 0.10 | 0.05 | 0.05 |

## 6 FROM DESCRIPTION TO PREDICTION

The proposed theoretical model makes a series of simplifications. We now demonstrate its predictive power of what actually happens in deep learning under specific interventions – the following experiments on GDE are conducted *after* we derived the theoretical framework and to the best of our knowledge have never been done in prior works. In other words, our model has not been specifically adjusted to account for the results of these experiments, but rather the experiments are designed based on the framework. Results for both experiments are shown in Table 1 (with uncertainty in Table 2) and the experimental details are in Appendix F.8 and F.9.

The first experiment considers **merging classes**. We observed in Section 5.2 that for GDE to hold approximately, the features distribution needs specific properties, namely, $t_r \propto t_d$ and $n_r \propto n_d$. If features do not interfere with each other significantly, our framework predicts that merging classes into superclasses should not change the ratios and thus would not break the GDE. We merge the classes of CIFAR 10 into different superclasses and run the same learning algorithms as Section 4 on the new data (6 random seeds). The accuracy-agreement difference does not change significantly across different partitions as predicted, even though the accuracy and agreement are different.

This result suggests that breaking the GDE requires intervening on the covariate distribution in order to change $\frac{t_r}{t_d}$. Thus, our second set of experiments considers **re-partitioning data**. In particular, we sort CIFAR 10 images by the proportion of blue in their total color intensity and partition them into 5 equally sized groups with increasing blue intensity. We observed that the accuracy-agreement differences of different data partitions are drastically different, corroborating the prediction made by the theoretical framework. Furthermore, we see that group 0 has the largest difference between accuracy and agreement which according to our theoretical framework suggests that the total number of rare features is larger. Through visual inspection (Figure 17), we can see that in group 1, the examples seem more visually complex and diverse, which could lead to a larger number of rare features (i.e., larger $t_r$). It is worth noting that, unlike the setting of Kirsch and Gal (2022), each group is still i.i.d. Therefore, the violation of GDE immediately implies that the ensemble is not calibrated on the data partition (Theorem 4.2 of Jiang et al. (2022)). We believe this is the first direct, non-adversarial construction of natural datasets where a deep ensemble is not well-calibrated in-distribution from a dataset on which the deep ensemble is usually well-calibrated.

## 7 CONCLUSION

We investigate distributions of features in data and how neural networks perform feature learning. Based on the empirical observations, we propose a new framework for understanding feature learning. We show that the proposed framework is more reflective of reality and can explain other phenomena in deep learning, notably GDE, without making any assumption about calibration. We believe this work provides new insight into our understanding of feature learning and data distribution in deep learning. The proposed framework could be useful for studying other phenomena related to agreement and ensembles such as calibration (Jiang et al., 2022), phenomena related to distribution shift such as accuracy-on-the-line (Miller et al., 2021) and agreement-on-the-line (Baek et al., 2022), and transfer learning. We discuss our framework's limitations and future directions in Appendix D.3. The new empirical tools we introduced can be valuable for other empirical investigations beyond the scope of this work. While recognizing that our framework, like all models, has its simplifications, we hope the intriguing results will catalyze more future works to study feature learning in this direction.

ACKNOWLEDGEMENT

We would like to thank Vaishnavh Nagarajan, Samuel Sokota, Elan Rosenfeld, Saurabh Garg, Jeremy Cohen, and Zixin Wen for the helpful discussion. We also thank Victor Akinwande, Zhili Feng, and Josh Williams for their feedback on an early draft of this work. Yiding and Christina were partially supported by funding from the Bosch Center for Artificial Intelligence. Yiding is also supported by the Google PhD fellowship.

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

# A  LIST OF NOTATIONS

## A.1  CONCEPTUAL FRAMEWORK

### A.1.1  MAIN PARAMETERS OF THE TOY MODEL

- $p_d$: the proportion of all data that are dominant.
- $p_r$: the proportion of all data that are rare. This parameter is equal to $1 - p_d$.
- $c$: the total model capacity. It represents how many features a single model can learn.
- $t_d$: the total number of dominant features available in the data for one class.
- $t_r$: the total number of rare features available in the data for one class.
- $n_d$: the total number of dominant features a single dominant data point has. $n_d \leq t_d$.
- $n_r$: the total number of rare features a single rare data point has. $n_r \leq t_r$.

### A.1.2  OTHER NOTATIONS OF THE TOY MODEL

- $\Psi(\cdot)$: a function that returns the set of features learned by a model $f$ or contained in a data point $x$. The intersection of $\Psi(\cdot)$ of different models and data determines how the model makes mistakes in our framework.
- $\text{err}(f, x, y)$: the error function for a model $f$ on the data $(x, y)$.
- $c_r$: The total capacity the model has for rare features. $c_r = \frac{1}{2}p_r c$.
- $c_d$: The total capacity the model has for dominant features. $c_r = \frac{1}{2}p_d c$.
- $\zeta$: the agreement function that dictates the probability with which two models that do not share features will agree.

## A.2  INTERACTION TENSOR

- $\Phi \in \mathbb{R}^{N \times d}$: Matrix that contains the representation of all data points for a single model.
- $\mathbf{V} \in \mathbb{R}^{d \times d}$: The principal components of $\Phi$.
- $\Phi_{\text{proj}} \in \mathbb{R}^{N \times k}$: Matrix that contains the projection of the representation of all data points.
- $\upsilon_{i,a}(x) \in \mathbb{R}$: the coefficient of the representation of $x$ in the $i^{\text{th}}$ model projected onto the $a^{\text{th}}$ principal component.
- $\mu_{i,a} \in \mathbb{R}$: the mean of $\upsilon_{i,a}(x)$ over the data distribution.
- $\sigma_{i,a}^2 \in \mathbb{R}$: the variance of $\upsilon_{i,a}(x)$ over the data distribution.
- $\rho_{(i,j),(a,b)} \in \mathbb{R}$: the Pearson correlation between $\upsilon_{i,a}(x)$ and $\upsilon_{j,b}(x)$.
- $\mathbf{K}_{i,j} \in [-1, 1]^{k \times k}$: the feature correlation matrix of two models $i$ and $j$.
- $\mathbf{\Lambda} \in [-1, 1]^{M \times M \times k \times k}$: the collection of all pairwise feature correlation matrices for all $M$ models. In other words, it contains the $\mathbf{K}_{i,j}$ for all pair of models.

# B  IRREDUCIBLE ERROR AND DATA SCALING

In this section, we discuss the sources of irreducible error in our framework. Concretely, there are two sources of irreducible error:

1. **Inductive bias mismatch**: these are errors that arise from the fact the models fundamentally *cannot* learn some of the features present in the data via conventional training, e.g., stochastic gradient descent.

2. **Finite sample error**: these are the errors that arise from insufficient sample size, where the models do not observe all the features in the data.

In both cases, the errors result from the models being unable to learn all the $2(t_d + t_r)$ features in the support of $\mathscr{D}$. We will refer to the percentage of all features that are present in the training data as *coverage* and use $\beta$ to denote it. When irreducible error occurs, in the best case, the best possible model can only learn up to $\beta_d t_d$ dominant features and $\beta_r t_r$ rare features for each class. Further, while we have previously assumed that there are always more features than the model capacity $c$, we will make the a mild but new assumption: *if the model's capacity is larger than the coverage, the model will sample noise for the remaining capacity.* Concretely, the model may be memorizing noise patterns in the data that do not help generalization similar to (Allen-Zhu and Li, 2020). We can characterize the expected accuracy when the irreducible error occurs (Proof in Appendix C.3).

**Lemma B.1.** *Under the proposed framework, with coverage of $\beta_d$ and $\beta_r$, the expected accuracy is upper-bounded by:*

$$\text{Acc} \leq p_d \left( 1 - \frac{1}{2} \frac{\binom{(1-\beta_d)t_d}{n_d}}{\binom{t_d}{n_d}} \right) + p_r \left( 1 - \frac{1}{2} \frac{\binom{(1-\beta_r)t_r}{n_r}}{\binom{t_r}{n_r}} \right). \tag{5}$$

Lemma B.1 provides an upper bound on the expected accuracy under this framework when the models cannot learn all the features. To test the validity of this hypothesis, we simulate different coverage by using training sets of different sizes. Specifically, we use training set size at 5% increment from 5% to 100% on CIFAR 10 and ResNet18. In Figure 5a, we show the the upperbound in Equation 5 as a function of coverage $\beta$ (same value for both $\beta_r$ and $\beta_d$). In Figure 5b, we show the test accuracy as the function of training set size.

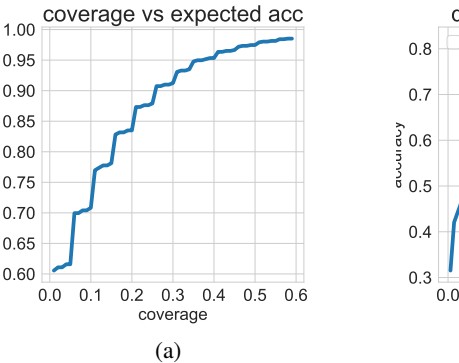

(a)  (b)

Figure 5: Plots of the predicted accuracy as a function of coverage (left) and real model accuracy as a function of the percent of training data (right).

Note that the lemma describes the *average-case test error* rather than the *worst-case test error* that classical bounds based on uniform convergence describe. Figure 5 shows that varying coverage can approximate the behavior of scaling dataset size (Hestness et al., 2017). Nonetheless, we see that some discrepancies between the two plots remain. Most notable is the fact that the test accuracy seems to increase at a faster rate than the accuracy described by the framework when the dataset is small. This difference exists likely because the relationship between training dataset size and coverage is *not linear*. In particular, coverage increases faster when the dataset size is small but saturates after the dataset size becomes large. One explanation for this phenomenon is that the models learn features differently in the presence of different dataset sizes. Our framework currently does not account for this effect but it is a promising direction for future works.

## C  FULL PROOF

In this section, we provide the full proof of the theoretical results. For convenience, we repeat the claims here.

## C.1 EXPECTED ACCURACY

**Proposition C.1.** *Under the proposed model, the expected accuracy over the model distribution and data distribution is:*

$$\text{Acc} = p_d \left( 1 - \frac{1}{2} \frac{\binom{t_d - c_d}{n_d}}{\binom{t_d}{n_d}} \right) + p_r \left( 1 - \frac{1}{2} \frac{\binom{t_r - c_r}{n_r}}{\binom{t_r}{n_r}} \right)$$

*Proof.* We are interested in computing the expected accuracy over the entire data distribution $\mathscr{D}$ *and* the entire hypothesis distribution $\mathscr{F}_\mathcal{A}$. It is simpler to think about the average error instead of accuracy (the prediction is random when the model and data do not share any features so the error is $\frac{1}{2}$):

$$\mathbb{E}_{f \sim \mathscr{F}_\mathcal{A}} \left[ \mathbb{E}_{(x,y) \sim \mathscr{D}} \left[ \text{err}(f, x, y) \right] \right] \tag{6}$$

$$= \mathbb{E}_{f \sim \mathscr{F}_\mathcal{A}} \left[ \mathbb{E}_{(x,y) \sim \mathscr{D}} \left[ \frac{1}{2} \mathbb{1} \left\{ \Psi(f) \cap \Psi(x) = \varnothing \right\} \right] \right] \tag{7}$$

$$= \frac{1}{2} \mathbb{P} \left[ \Psi(f) \cap \Psi(x) = \varnothing \right] \tag{8}$$

$$= \frac{1}{2} \mathbb{P} \left[ \text{ a sampled } f \text{ and a sampled } x \text{ do not share any features} \right] \tag{9}$$

To avoid notational clutter, we will use $\mathbf{f}$ and $\mathbf{x}$ to denote *a sampled* $f$ and *a sampled* $x$. The probability of interest is thus:

$$\mathbb{P} \left[ \mathbf{f} \text{ and } \mathbf{x} \text{ do not share any features} \right] = 1 - \mathbb{P} \left[ \mathbf{f} \text{ and } \mathbf{x} \text{ share at least 1 features} \right] \tag{10}$$

We first partition the event space into two parts: 1. $\mathbf{x}$ is dominant, and 2. $\mathbf{x}$ is rare. Suppose that $\mathbf{x}$ is dominant, we want to compute $\mathbb{P} \left[ \mathbf{f} \text{ and } \mathbf{x} \text{ do not share any features} \mid \mathbf{x} \text{ is dominant} \right]$. Since all $f$'s have an equal probability of being sampled (as they contain the same numbers of problematically indistinguishable features), the probability is equivalent to:

$$\mathbb{P} \left[ \mathbf{f} \text{ and } \mathbf{x} \text{ do not share any features } \mid \mathbf{x} \text{ is dominant} \right] \tag{11}$$

$$= \mathbb{P} \left[ \mathbf{x} \text{ does not have a fixed set of } c_d \text{ features } \mid \mathbf{x} \text{ is dominant} \right] \tag{12}$$

$$= \frac{\binom{t_d - c_d}{n_d}}{\binom{t_d}{n_d}}. \tag{13}$$

This is the configuration of $\mathbf{x}$ that does not contain the $c_d$ features of $f$. Analogously, we can compute:

$$\mathbb{P} \left[ \mathbf{f} \text{ and } \mathbf{x} \text{ do not share any features } \mid \mathbf{x} \text{ is rare} \right] = \frac{\binom{t_r - c_r}{n_r}}{\binom{t_r}{n_r}}.$$

Given the assumption about how models make mistakes, the expected for both parts of the event space:

$$\text{Acc}_d = \frac{1}{2} \cdot \mathbb{P} \left[ \mathbf{f} \text{ and } \mathbf{x} \text{ do not share any features } \mid \mathbf{x} \text{ is dominant} \right]$$
$$+ 1 \cdot \mathbb{P} \left[ \mathbf{f} \text{ and } \mathbf{x} \text{ share at least 1 features } \mid \mathbf{x} \text{ is dominant} \right] \tag{14}$$

$$= \frac{1}{2} \frac{\binom{t_d - c_d}{n_d}}{\binom{t_d}{n_d}} + 1 - \frac{\binom{t_d - c_d}{n_d}}{\binom{t_d}{n_d}} = 1 - \frac{1}{2} \frac{\binom{t_d - c_d}{n_d}}{\binom{t_d}{n_d}}, \tag{15}$$

Analogously we repeat the computation for rare data,

$$\text{Acc}_r = 1 - \frac{1}{2} \frac{\binom{t_r - c_r}{n_r}}{\binom{t_r}{n_r}}. \tag{16}$$

We now compute the expected accuracy over the entire event space,

$$\text{Acc} = p_d \text{Acc}_d + p_r \text{Acc}_r \tag{17}$$

$$= p_d \left( 1 - \frac{1}{2} \frac{\binom{t_d - c_d}{n_d}}{\binom{t_d}{n_d}} \right) + p_r \left( 1 - \frac{1}{2} \frac{\binom{t_r - c_r}{n_r}}{\binom{t_r}{n_r}} \right) \tag{18}$$

$\square$

## C.2 Expected Agreement

**Proposition C.2.** *Under the proposed model, let $\binom{n}{r} = 0$ when $n < 0$, $r < 0$ or $n < r$, and further define:*

$$q_1 = p_d \left( 1 - \frac{\binom{t_h - c_d}{n_d}}{\binom{t_d}{n_d}} \right)^2 + p_r \left( 1 - \frac{\binom{t_r - c_r}{n_r}}{\binom{t_r}{n_r}} \right)^2,$$

$$q_2(k) = \quad p_d \frac{\binom{t_d - n_d}{c_d}^2}{\binom{t_d}{c_d}^2} \left( \sum_{a+b=k} \frac{\binom{c_d}{a}\binom{t_d - n_d - c_d}{c_d - a}}{\binom{t_d - n_d}{c_d}} \frac{\binom{c_r}{b}\binom{t_r - c_r}{c_r - b}}{\binom{t_r}{c_r}} \right)$$

$$+ p_r \frac{\binom{t_r - n_r}{c_r}^2}{\binom{t_r}{c_r}^2} \left( \sum_{a+b=k} \frac{\binom{c_d}{a}\binom{t_d - c_d}{c_d - a}}{\binom{t_d}{c_d}} \frac{\binom{c_r}{b}\binom{t_r - n_r - c_r}{c_r - b}}{\binom{t_r - n_r}{c_r}} \right),$$

$$q_3 = 1 - q_1 - \sum_{k=1}^{c} q_2(k),$$

*the expected agreement between an i.i.d pair of model $(f, g)$ drawn for the model distribution over the data distribution is:*

$$\mathsf{Agr} = q_1 + \frac{1}{2}q_3 + \sum_{k=1}^{c} \zeta(k, c) q_2(k)$$

*Proof.* We are interested computing the expected disagreement of $(f, g) \sim \mathscr{F}_A \times \mathscr{F}_A$ over the data distribution $x \sim \mathscr{D}$. Based on the features in $f$ and $g$, We partition the event space into 3 subsets:

- A: $f$ and $g$ both share features with $x$.

$$\left\{ \Psi(f) \cap \Psi(x) \neq \varnothing \right\} \bigcap \left\{ \Psi(g) \cap \Psi(x) \neq \varnothing \right\}$$

- B: $f$ and $g$ do not share any features with $x$ but share features with each other.

$$\left\{ |\Psi(f) \cap \Psi(g)| \neq \varnothing \right\} \bigcap \left\{ \Psi(f) \cap \Psi(x) = \varnothing \right\} \bigcap \left\{ \Psi(g) \cap \Psi(x) = \varnothing \right\}$$

- C: The rest of the event space. In these events, we have either:
  - $f$ and $g$ do not share any features with $x$ or each other.
  - only one of $f$ and $g$ share features with $x$.

**Case A.** Since $f$ and $g$ are independent and identically distributed, it suffices to compute the probability of one of them not sharing any features with $x$. We further partition the event space into two part conditioned on whether the data point is dominant or rare (this is possible because $x$ is independent from $f$ and $g$). Following the same logic as Equation 11:

$$\mathbb{P}[\,\Psi(f) \cap \Psi(\mathbf{x}) \neq \varnothing \mid \mathbf{x} \text{ is dominant}\,] \tag{19}$$

$$= 1 - \mathbb{P}[\Psi(f) \cap \Psi(\mathbf{x}) = \varnothing \mid \mathbf{x} \text{ is dominant}\,] \tag{20}$$

$$= 1 - \frac{\binom{t_d - c_d}{n_d}}{\binom{t_d}{n_d}} \tag{21}$$

Analogously,

$$\mathbb{P}[\Psi(f) \cap \Psi(\mathbf{x}) \neq \varnothing \mid \mathbf{x} \text{ is rare}\,] = 1 - \frac{\binom{t_r - c_r}{n_r}}{\binom{t_r}{n_r}}. \tag{22}$$

By the independence of $f$ and $g$:

$$\mathbb{P}[\, A \mid \mathbf{x} \text{ is dominant} \,] \tag{23}$$

$$= \mathbb{P}[\Psi(f) \cap \Psi(\mathbf{x}) \neq \varnothing \mid \mathbf{x} \text{ is dominant}]^2 \tag{24}$$

$$= \left( 1 - \frac{\binom{t_d - c_d}{n_d}}{\binom{t_d}{n_d}} \right)^2, \tag{25}$$

and similarly,

$$\mathbb{P}[A \mid \mathbf{x} \text{ is rare}] = \left( 1 - \frac{\binom{t_r - c_r}{n_r}}{\binom{t_r}{n_r}} \right)^2. \tag{26}$$

Putting everything together:

$$q_1 = \mathbb{P}[A] = p_d \left( 1 - \frac{\binom{t_r - c_r}{n_r}}{\binom{t_r}{n_r}} \right)^2 + p_r \left( 1 - \frac{\binom{t_r - c_r}{n_r}}{\binom{t_r}{n_r}} \right)^2. \tag{27}$$

**Case B.**  Again, we partition the event space based on dominant and rare data. Then we further partition the even space based on $k$, the number of features that $f$ and $g$ share with each other. First, we compute the probability that both $f$ and $g$ do not share any features with $x$. By independence and equation 11:

$$\mathbb{P}[\Psi(f) \cap \Psi(\mathbf{x}) = \varnothing \ \text{ and } \ \Psi(g) \cap \Psi(\mathbf{x}) = \varnothing \mid \mathbf{x} \text{ is dominant}] \tag{28}$$

$$= \mathbb{P}[\Psi(f) \cap \Psi(\mathbf{x}) = \varnothing \mid \mathbf{x} \text{ is dominant}]^2 \tag{29}$$

$$= \frac{\binom{t_d - c_d}{n_d}^2}{\binom{t_d}{n_d}^2}. \tag{30}$$

Conditioned on that $x$ is dominant and that $f$ and $g$ do not share any features with $x$, we now compute the probability where $f$ and $g$ share exactly $k$ features. By symmetry, this probability is equal to the probability of sampling $g$ that shares exactly $k$ features with a *fixed f*. Since $f$ and $g$ cannot share any feature with $x$, the total number of dominant features available is $t_d - n_d$. This event space can be further partitioned into disjoint events where $g$ shares exactly $a$ dominant features and $b$ rare features with $f$ for $(a, b) \in \{(0, k), (1, k-1), \ldots, (k-1, 1), (k, 0)\}$. Since $g$ always samples $c_d$ dominant features and $c_r$ rare features, the two processes are independent from each other and respectively follow hypergeometric distributions (i.e., marble picking problem):

$$\mathbb{P}\left[ |\Psi(\mathbf{f}) \cap \Psi(\mathbf{g})| = k \mid \Psi(\mathbf{f}) \cap \Psi(\mathbf{x}) = \varnothing \ \text{ and } \ \Psi(\mathbf{g}) \cap \Psi(\mathbf{x}) = \varnothing \ \text{ and } \ \mathbf{x} \text{ is dominant} \right]$$

$$= \sum_{a+b=k} \frac{\binom{c_d}{a} \binom{t_d - n_d - c_d}{c_d - a}}{\binom{t_d - n_d}{c_d}} \frac{\binom{c_r}{b} \binom{t_r - c_r}{c_r - b}}{\binom{t_r}{c_r}}.$$

The first term in the summation is the density of the hypergeometric distribution for sampling $a$ allowed dominant features, and the second term is the hypergeometric distribution for sampling $b$ allowed rare features.

The same reasoning process can be applied to when $\mathbf{x}$ is rare by modifying the available number of rare features to $t_r - n_r$ and keep the available number of dominant features as $t_d$:

$$\mathbb{P}[\Psi(f) \cap \Psi(\mathbf{x}) = \varnothing \ \text{ and } \ \Psi(g) \cap \Psi(\mathbf{x}) = \varnothing \mid \mathbf{x} \text{ is rare}] = \frac{\binom{t_r - c_r}{n_r}^2}{\binom{t_r}{n_r}^2} \tag{31}$$

$$\mathbb{P}\left[ |\Psi(\mathbf{f}) \cap \Psi(\mathbf{g})| = k \mid \Psi(\mathbf{f}) \cap \Psi(\mathbf{x}) = \varnothing \ \text{ and } \ \Psi(\mathbf{g}) \cap \Psi(x) = \varnothing \ \text{ and } \ \mathbf{x} \text{ is rare} \right] \tag{32}$$

$$= \sum_{a+b=k} \frac{\binom{c_d}{a} \binom{t_d - c_d}{c_d - a}}{\binom{t_d}{c_d}} \frac{\binom{c_r}{b} \binom{t_r - n_r - c_r}{c_r - b}}{\binom{t_r - n_r}{c_r}}. \tag{33}$$

Putting everything together, we arrive at the probability:

$$q_2(k) = \mathbb{P}[|\Psi(\mathbf{f}) \cap \Psi(\mathbf{g})| = k \text{ and } \Psi(\mathbf{f}) \cap \Psi(\mathbf{x}) = \varnothing \text{ and } \Psi(\mathbf{g}) \cap \Psi(\mathbf{x}) = \varnothing] \tag{34}$$

$$= p_d \frac{\binom{t_d-n_d}{c_d}^2}{\binom{t_d}{c_d}^2} \left( \sum_{a+b=k} \frac{\binom{c_d}{a}\binom{t_d-n_d-c_d}{c_d-a}}{\binom{t_d-n_d}{c_d}} \frac{\binom{c_r}{b}\binom{t_r-c_r}{c_r-b}}{\binom{t_r}{c_r}} \right) \tag{35}$$

$$+ p_r \frac{\binom{t_r-n_r}{c_r}^2}{\binom{t_r}{c_r}^2} \left( \sum_{a+b=k} \frac{\binom{c_d}{a}\binom{t_d-c_d}{c_d-a}}{\binom{t_d}{c_d}} \frac{\binom{c_r}{b}\binom{t_r-n_r-c_r}{c_r-b}}{\binom{t_r-n_r}{c_r}} \right). \tag{36}$$

Note that there may be cases where the combination is undefined (e.g., $t_d - n_d - c_d < 0$ or $t_d - n_d - c_d < c_d - a$). These cases means that the configurations are impossible to exist, so their corresponding probabilities are 0. We will define $\binom{n}{r} = 0$ when $n < 0$, $r < 0$ or $n < r$ to handle these cases. The total probability of $B$ is equal to the sum of $q(k)$ from $k = 1$ to $c$ since that is equivalent of the event $|\Psi(f) \cap \Psi(g)| > 0$:

$$\mathbb{P}[B] = \sum_{k=1}^{c} q_2(k) \tag{37}$$

**Case C.** This event is the complement of $A \cup B$ so:

$$q_3 = \mathbb{P}[C] = 1 - \mathbb{P}[A] - \mathbb{P}[B] = 1 - q_1 - \sum_{k=1}^{c} q_2(k). \tag{38}$$

In A, we know the models agree with probability 1. In C, either both models will make a random guess or one model will make a random guess and the other will classify $x$ correctly. In both cases, they will agree with probability $\frac{1}{2}$. In B, we assumed that the probability agreement is modulated by the agreement function $\zeta$ (Section 5). Combining these agreement conditions with the probability of A, B, C gives:

$$\mathsf{Agr} = 1 \cdot \mathbb{P}[A] + \frac{1}{2} \cdot \mathbb{P}[C] + \sum_{k=1}^{c} q_2(k)\zeta(k,c) \tag{39}$$

$$= q_1 + \frac{1}{2}q_3 + \sum_{k=1}^{c} q_2(k)\zeta(k,c). \tag{40}$$

Replacing $q_3$ with $1 - q_1 - \sum_{k=1}^{c} q_2(k)$ and simplify yields the final results.

$\square$

## C.3 COVERAGE LEMMA

**Lemma C.3.** *Under the proposed framework, with coverage of $\beta_d$ and $\beta_r$, the expected accuracy is upper-bounded by:*

$$\mathsf{Acc} \leq p_d \left( 1 - \frac{1}{2} \frac{\binom{(1-\beta_d)t_d}{n_d}}{\binom{t_d}{n_d}} \right) + p_r \left( 1 - \frac{1}{2} \frac{\binom{(1-\beta_r)t_r}{n_r}}{\binom{t_r}{n_r}} \right). \tag{41}$$

*Proof.* Lets call the set of all features $\Gamma$ and set of features available for the models to learn $\widehat{\Gamma}$. We can naturally partition them based on dominant and rare features – $\Gamma_d$ is the set of all dominant features and $\Gamma_r$ is the set of all rare features. By the coverage assumption $|\widehat{\Gamma}_r| = \beta_r|\Gamma_r|$ and $|\widehat{\Gamma}_d| = \beta_d|\Gamma_d|$.

Notice that having different numbers of features available to the models and data means that the distributions of model sharing features with conditioned on the data is no longer the identical for different data. The conditional probability changes depending on how many features of the data point is not in $\widehat{\Gamma}$. On the other hand, conditional probability of data point sharing features with a fixed model is the same for all models, because $\widehat{\Gamma} \subseteq \Gamma$ — No matter what features are in $\Psi(f)$, the

probability that a sampled data point does not share any dominant features with it is $\binom{t_d-c_d}{n_d}/\binom{t_d}{n_d}^3$.
Recall that $c_d$ and $c_r$ represent how many features the model can learn which is upperbounded by $\beta_d t_d$ and $\beta_r t_r$. Since $\binom{n}{r}$ is monotonically increasing in $n$:

$$\beta_d t_d \geq c_d \implies t_d - \beta_d t_d \leq t_d - c_d \implies \binom{(1-\beta_d)t_d}{n_d} \leq \binom{t_d - c_d}{n_d}, \tag{42}$$

the same can be derived for rare data. Substituting in the expression for accuracy from Equation 3,

$$\text{Acc} = p_d\left(1 - \frac{1}{2}\frac{\binom{t_d-c_d}{n_d}}{\binom{t_d}{n_d}}\right) + p_r\left(1 - \frac{1}{2}\frac{\binom{t_r-c_r}{n_r}}{\binom{t_r}{n_r}}\right) \tag{43}$$

$$\leq p_d\left(1 - \frac{1}{2}\frac{\binom{(1-\beta_d)t_d}{n_d}}{\binom{t_d}{n_d}}\right) + p_r\left(1 - \frac{1}{2}\frac{\binom{(1-\beta_r)t_r}{n_r}}{\binom{t_r}{n_r}}\right). \tag{44}$$

$\square$

# D  FURTHER DISCUSSIONS OF THE THEORETICAL MODEL

## D.1  COMPARISON TO PRIOR AND CONCURRENT WORKS

An important difference between this model and the multi-view model from Allen-Zhu and Li (2020) is that our model does not treat all features as having the same learning difficulty (i.e., probability of being learned). Indeed, the experiments in Section 4 show that features demonstrate a wide range of behaviors in terms of how often they occur in the data and how they interact with the models. Another notable difference is that in Allen-Zhu and Li (2020), the multi-view portion of the dataset contains *all* the features. In reality, the "easy" part of the data that a large portion of the models classifies correctly actually contains much fewer features. These observations suggest that having different types of features may be a more accurate description of nature. Nonetheless, we do not describe the exact mechanism of how feature learning actually happens under our model since we are not assuming any particular hypothesis class. Consequently, we do not use the same definition as Allen-Zhu and Li (2020) as they adopt a very simplified model of features (i.e., orthogonal vectors in the input space). The spirit of our model of feature learning is close to that of Allen-Zhu and Li (2020) and we believe a similar iterative analysis can be applied to our model.

There is also a recent line of theoretical works that depart from the random feature model and try to incorporate feature learning into the analysis (Damian et al., 2022; Wang et al., 2022; Moniri et al., 2023; Nichani et al., 2023; Dandi et al., 2023). These works generally assume the data is an isotropic Gaussian and the target function is a 2-layer neural network. The feature is usually a linear function of the input and sometimes special training algorithms are needed (e.g., layer-wise training). A notable development is Nichani et al. (2023) which analyzed 3-layer neural network which learns *non-linear* features although it still requires layer-wise training. In general, there is still a gap between the assumptions made in these works and real models and, to the best of our knowledge, they do not account for ensemble or the possibility of learning different features. The goal of our work is not to discuss the merits or limitations of these works but to study what actually happens in practice on real dataset and real architectures. Due to the complex nature of deep learning, we believe this more scientific approach is a good complement to the existing theoretical works.

It is also natural to question whether the simplification where a single feature is sufficient for determining the class is sensible. We believe that this simplification is realistic for a binary classification problem and that using more features in determining the true class may make the model more expressive but should not fundamentally alter the behavior of the system. Further, the true data distributions are evidently more complex — dominant data can contain rare features, and, vice versa. In fact, both features and data can lie on a continuous spectrum between "dominant" and "rare" (Figure 2b and 2c). These changes can be incorporated into the framework by modifying the distribution of features but doing so can increase the complexity of the analysis and require tail-bounds to characterize the system's behavior.

---

[3]Here we assume the capacity is smaller than the number of available features. If the capacity is larger, then model will learn all available features and the bound is tight.

### D.2 Sources of randomness

Another assumption we made is that when the model $f$ does not share any feature with a data point $x$, the model will make a random guess. At first look, this seems like a strong assumption that requires the model to make a perfectly random guess. However, recall that we are computing the expectation over the model distribution and the data distribution rather than a single fixed data point. For a single model $f$, its prediction is effectively random if its average prediction over all the distribution of data that do not share features with $f$ is at the chance:

$$\mathbb{P}_{\mathscr{D}}[f(\mathbf{x}) = \mathbf{y} \mid \Psi(f) \cap \Psi(\mathbf{x}) = \varnothing] = \mathbb{E}_{(\mathbf{x},\mathbf{y}) \sim \mathscr{D}}\left[\mathbb{1}\{f(\mathbf{x}) = \mathbf{y}\} \mid \Psi(f) \cap \Psi(\mathbf{x}) = \varnothing\right] = \frac{1}{2}. \quad (45)$$

This means that $f$ can be *completely deterministic* as long as its accuracy over all the data that it doesn't share feature with is random chance. This is in fact the only sensible outcome if we assume that features are indeed what the models use to make predictions. In this case, *the source of randomness comes from the data*, $(\mathbf{x}, \mathbf{y}) \sim \mathscr{D}$.

We now analyze the case where we hold a single data point $(x, y)$ fixed and generate the source of randomness from the training algorithm $f \sim \mathscr{F}_{\mathcal{A}}$ (once again, the individual model can be completely deterministic). When the data point is one with which $f$ does not share features, we cannot expect the models to make independent predictions since the models have similar inductive bias and can make predictions in a correlated manner depending on $x$ (e.g., noise in $x$):

$$\mathbb{P}_{\mathscr{F}_{\mathcal{A}}}[\mathbf{f}(x) = y \mid \Psi(\mathbf{f}) \cap \Psi(x) = \varnothing] = \mathbb{E}_{\mathbf{f} \sim \mathscr{F}_{\mathcal{A}}}\left[\mathbb{1}\{\mathbf{f}(x) = y\} \mid \Psi(\mathbf{f}) \cap \Psi(x) = \varnothing\right] \neq \frac{1}{2}. \quad (46)$$

Consequently, the agreement between a pair of models will not be random over the data distribution, and this is exactly what the agreement function tries to model.

$$\mathbb{P}_{\mathscr{F}_{\mathcal{A}} \times \mathscr{F}_{\mathcal{A}}}[\mathbf{f}(x) = \mathbf{g}(x) \mid \Psi(\mathbf{f}) \cap \Psi(x) = \varnothing \text{ and } \Psi(\mathbf{g}) \cap \Psi(x) = \varnothing] = \zeta(\mathscr{F}_{\mathcal{A}}, x) \quad (47)$$

In the most general case, $\zeta$ is a function of the hypothesis distribution and a data point $x$, but the ones we used in the main text assume that $\zeta$ is a function of the model's features, since what type of data $x$ is irrelevant if neither models have the features to predict it so we can also drop that dependency.

### D.3 Limitations

While our theoretical framework is able to explain some previously poorly understood phenomena, several limitations still exist. We have discussed various limitations of our framework throughout the paper but for ease of reading, we will discuss all of them here.

**Optimization analysis**   One of the important limitations of this work is that it only describes what kind of features are present in the data and end up being learned by different models, but it does not offer any insight on *how* these features are learned through gradient descent mechanistically. Modeling the optimization procedure is evidently a challenging task (although this is true for deep learning in general). One promising avenue is via techniques similar to Allen-Zhu and Li (2020).

**Binary dichotomy of features**   The current conceptual model assumes there are only two types of features, which facilitate the analysis. Evidently, Figure 2b shows that the features in fact live on a spectrum of frequency of appearance. Future work could resolve this by assuming long-tailed distributions (e.g., Zipf's distribution) instead of binary distributions to make the model closer to reality.

**Binary classification**   The current conceptual model only analyzes a binary classification. We discuss steps towards extending it to multi-class in Appendix D.4.

### D.4 Extension to multi-class

In order to extend this framework to multi-class, we would first have to decide on how the model makes predictions based on the features it has learned and the features present in the data. In this setting, perfect prediction based on a single feature may no longer be enough since different classes can share features. Instead, one may need to introduce a new function for the probability of correct classification based on the number of shared features between the model and the data point or the

probability of making a mistake based on the features. This also means that we cannot no longer assume the model will make a random guess since there are more than one possible wrong class and how the model makes a prediction will depend on the features they share with these wrong classes. Mathematically, this means that $\zeta$ is no longer independent of the data point $x$. The desired quantities are still computable through combinatorics but the added complexity could make the derivation much more complicated and an analytical expression may or may not be attainable, though the problem may be amenable through tail-bounds.

# E    EXPERIMENTAL DETAILS

In our experiments, we use two collections of $k = 20$ ResNet18 (He et al., 2016a) trained on the CIFAR-10 dataset (Krizhevsky et al., 2009) following the experimental set up of Jiang et al. (2022). The first collection of models is trained on random 10000 subsets of the whole training set (`10k`), and the second collection of models is trained on random 45000 subsets of the whole training set (`45k`). On average, the `10k` models achieve $67.9\%$ test accuracy and the `45k` models achieve $84.5\%$ test accuracy. In addition, we repeat the same process for SVHN dataset Netzer et al. (2011) using random 45000 subsets of the whole training set (`SVHN`). The training details are outlined in Appendix E and E.2. We compute $\boldsymbol{\Lambda}$ and $\boldsymbol{\Omega}$ on the test set using the output of the penultimate layer (i.e., $\psi$ is the final linear layer). For clustering features, we choose $k = 50$ for the number of principal components to use, $\gamma_{\mathrm{corr}}$ to be the $90^{\mathrm{th}}$ percentile of $\boldsymbol{\Lambda}$, and $\gamma_{\mathrm{data}}$ to be the $90^{\mathrm{th}}$ percentile of all entries in $\widehat{\Phi}_i^{\mathrm{proj}}$, $i = 1, \ldots, M$. After clustering, we obtain $T = 680$ feature clusters for `45k`. We primarily show `45k` here and leave the `10k` and `SVHN` results to Appendix F, which are similar to the `45k` results qualitatively.

## E.1    TRAINING DETAILS

For the ResNet18 experiments, we follow the same procedures as Jiang et al. (2022) which uses the same architecture of ResNet18 as He et al. (2016a). We train the 20 models with:

- initial learning rate: 0.1

- weight decay: 0.0001

- minibatch size: 100

- data augmentation: No

The models in `45k` samples 45000 data points from the training set without replacement. Likewise, the models in `10k` samples 10000 data points from the training set without replacement.

## E.2    CLUSTERING ALGORITHM

Algorithm 1 iterates over all entries of $\boldsymbol{\Lambda}$ and assigns each feature to a cluster if its correlation with the members of the cluster exceeds $\gamma_{\mathrm{corr}}$; otherwise, the algorithm creates a new cluster for that feature. One notable property of the greedy clustering algorithm is that it does not generate a fixed number of clusters. This is desirable in this case because if a feature does not have a high correlation with any other features, we would like to isolate it as a unique cluster rather than grouping it together with other features.

For the number of principal components, we recommend picking the number where after projecting every activation vector onto the principal components, the linear layer can classify the projected representation with approximately the same accuracy as the representation before projection. In our setting, 50 principal components could retain 100% of the original performance. For $\gamma_{\mathrm{corr}}$ , we found that the qualitative results are not very sensitive to different values. We experiment with different values of $\gamma_{\mathrm{corr}}$ in Appendix F.10 and observed similar results.

## E.3    HARDWARE

All experiments in the paper are done on single Nvidia RTX 2080's and RTX A6000's.

---

**Algorithm 1** ClusterFeatures

---

1: **Input**: $\mathbf{\Lambda}$[M,M,k,k], $\gamma_{\text{corr}}$
2: Assignment[M,k] ← new empty matrix
3: Maximum[M,k] ← new matrix filled with $-1$
4: CurrentFeature ← 1
5: **for** i = 1 to k and j = 1 to m **do**
6:     **if** Assignment[i,j] is not empty **then**
7:         Skip to the next j
8:     Assignment[i,j] ← CurrentFeature
9:     **for** p = 1 to k **do**
10:         CorrMat ← $\mathbf{\Lambda}$[i,p,:,:]
11:         FeatureRow ← CorrMat[j,:]
12:         **for** q = 1 to m **do**
13:             **if** FeatureRow[q] > Maximum[p,q] and FeatureRow[q] > $\gamma_{\text{corr}}$ **then**
14:                 Assignment[p,q] ← CurrentFeature
15:                 Maximum[p,q] ← FeatureRow[q]
16:     CurrentFeature ← CurrentFeature + 1
17: **Return** Assignment

---

# F   ADDITIONAL FIGURES, SIMULATIONS, AND EXPERIMENTS

## F.1   INTERACTION TENSOR

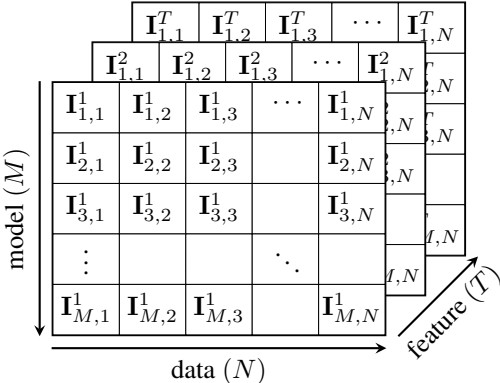

Figure 6: An illustration of the interaction tensor, $\mathbf{\Omega}$. The three axes correspond to *model*, *data*, and *features*. An entry $\mathbf{I}_{m,n}^t$ is 1 if both data $n$ and model $m$ contains feature $t$ and is 0 otherwise.

## F.2   EFFECT OF PCA

In Figure 8, we show the correlation matrices, $\mathbf{K}_{i,j}$, for different model pairs from 10k and 45k. The first column shows the self-correlation matrix between the features of the same model. Both matrices are effectively diagonal which indicates that the principal components represent features with *no redundant information*. This contrasts with Li et al. (2015) where the self-correlation matrices have many off-diagonal entries. Off-diagonal entries for the self-correlation matrix indicate that either there is redundant information or a single feature is distributed across multiple neurons, which is not desirable for studying unique features. Another interesting effect of using PCA projected features is that the features are naturally "aligned" because the principal components are already sorted by the amount of variance they can explain. We can see that the correlation matrices' entries (especially towards the top features) are naturally more concentrated towards the diagonal. Furthermore, the models with more data and higher test accuracy (45k) have more near diagonal entries. This indicates that the models in 45k have learned nearly the same top features. This observation is consistent with Li et al. (2015); Morcos et al. (2018) which find that better models tend to learn more similar representations.

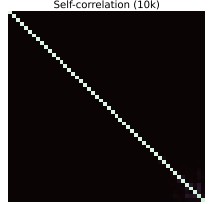 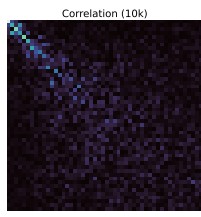 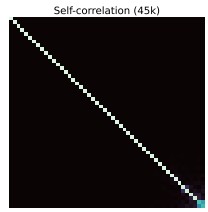 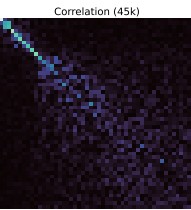

Figure 8: Self-correlation ($\mathbf{K}_{i,j}$ where $i = j$) and correlation ($\mathbf{K}_{i,j}$ where $i \neq j$) for random models from `10k` and `45k`. Both self-correlation matrices contain only diagonal entries, indicating that the features of the same model contain no redundant information. On the other hand, the correlation between models in `45k` exhibits more structure than the models in `10k`, with the non-zero entries more concentrated around the diagonal (zoom in for better visuals). This suggests that the features of models learned in `45k` contain more similar information compared to models learned in `10k`.

| | $\widetilde{K} = 2$ | $\widetilde{K} = 3$ | $\widetilde{K} = 5$ | $\widetilde{K} = 10$ | $G_1$ | $G_2$ | $G_3$ | $G_4$ | $G_5$ |
|---|---|---|---|---|---|---|---|---|---|
| `Acc` | $0.80_{\pm 0.01}$ | $0.84_{\pm 0.01}$ | $0.78_{\pm 0.01}$ | $0.81_{\pm 0.02}$ | $0.62_{\pm 0.02}$ | $0.60_{\pm 0.01}$ | $0.59_{\pm 0.03}$ | $0.66_{\pm 0.02}$ | $0.70_{\pm 0.03}$ |
| `Agr` | $0.85_{\pm 0.06}$ | $0.88_{\pm 0.05}$ | $0.83_{\pm 0.08}$ | $0.85_{\pm 0.07}$ | $0.70_{\pm 0.13}$ | $0.67_{\pm 0.15}$ | $0.69_{\pm 0.15}$ | $0.71_{\pm 0.13}$ | $0.75_{\pm 0.11}$ |
| `Diff` | 0.05 | 0.04 | 0.04 | 0.05 | 0.08 | 0.07 | 0.10 | 0.05 | 0.05 |

Table 2: Average accuracy and agreement on datasets with different interventions. $\widetilde{K}$ represents different number of superclasses and $\widetilde{K} = 10$ is the original CIFAR10. $G_i$ presents the data in the $(i-1) \times 20^{\text{th}}$ to $i \times 20^{\text{th}}$ percentile of blue intensity. The difference between accuracy and agreement is approximately the same for different $\widetilde{K}$'s (thus GDE holds) but not for different $G_i$'s. The uncertainty denotes standard deviation.

### F.3 EMPIRICAL PROPERTIES OF PCA FEATURES

**Features are semantically meaningful** As shown in Figure 1, our feature definitions are semantically meaningful and can be used for identifying common prototypes and rare images in each class. new experiments on the density of feature in each group? Rare images contain more rare features. This property of the defined features also allows us to find semantically similar images in the dataset. To do so, we first define a similarity metric between two images:

$$s(\boldsymbol{x}_1, \boldsymbol{x}_2) = \frac{2 \, |\Psi(\boldsymbol{x}_1) \cap \Psi(\boldsymbol{x}_2)|}{|\Psi(\boldsymbol{x}_1)| + |\Psi(\boldsymbol{x}_2)|}. \tag{48}$$

This function intuitively computes the overlap of features between two images normalized by their total number of features. For any given image $\boldsymbol{x}$, we can compute the similarity of $\boldsymbol{x}$ and the entire dataset and find the ones with the highest similarities. In Figure 9, we show the nearest neighbors of a random sample of images. We can see that our metric is able to identify semantically similar neighbors for each image even if the images are not close in pixel space.

Note that for the second row of Figure 9, the first 7 neighbors have 100% feature overlap. The property of our definition of feature may be of independent interest to other applications.

**Individual features do not correspond to particular classes** It may be tempting to think that individual features may correspond to individual classes. In the extreme case, this would reduce to neural collapse (Papyan et al., 2020) (which only happens after the model has been trained for an extremely long time). We find that this is not the case. Instead, individual features do not correspond to any particular classes (Figure 10). To illustrate this point further, we plot the frequency at which the top features appear in each class, and observed that the dominant features often appear in many different classes with different frequencies and would be missing from only one or two classes (Figure 11). This suggests that individual features can represent multiple "concepts" in the data but combinations of several features are much more interpretable (Figure 9).

This is perhaps not too surprising since in general we cannot expect the models to learn features that humans consider to be good features. After all, the appeal for using neural networks is the difficulty of designing hand-engineered features. Future works could investigate these observations further.

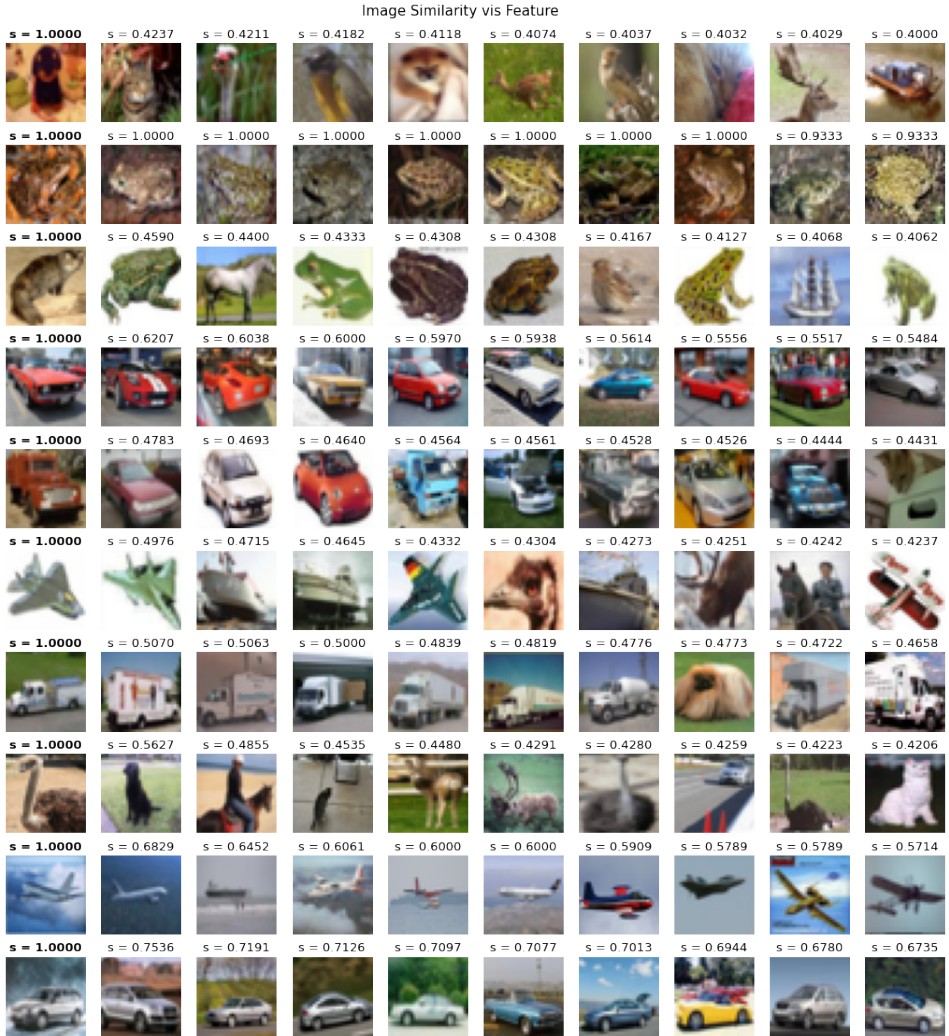

Figure 9: Nearest neighbors of random images measured by feature overlap. The leftmost image is the base image and the row contains its nearest neighbors. The similarity score is shown above each image. We can see that feature overlap can reliable capture the semantic similarities between different images even if the distance in pixels space is large. This phenomenon is particularly obvious in the second row where we can see that several distinct images of frogs have 100% feature overlap.

**Features capture prototypical examples.** We also observed (Figure 12) that our definition of features can recover the notion of *prototypical examples* observed in Carlini et al. (2019); Jiang et al. (2020). In particular, the images with the least features seem to correspond to the prototypical examples (images where the objects are presented in a canonical way) whereas the images with the most features seem to correspond to non-prototypical examples (images where the objects are presented in a rare way). This means that these prototypical examples usually contain much fewer (dominant) features whereas the non-prototypical examples contain much more rare features. Exploring these connections would be an interesting future direction.

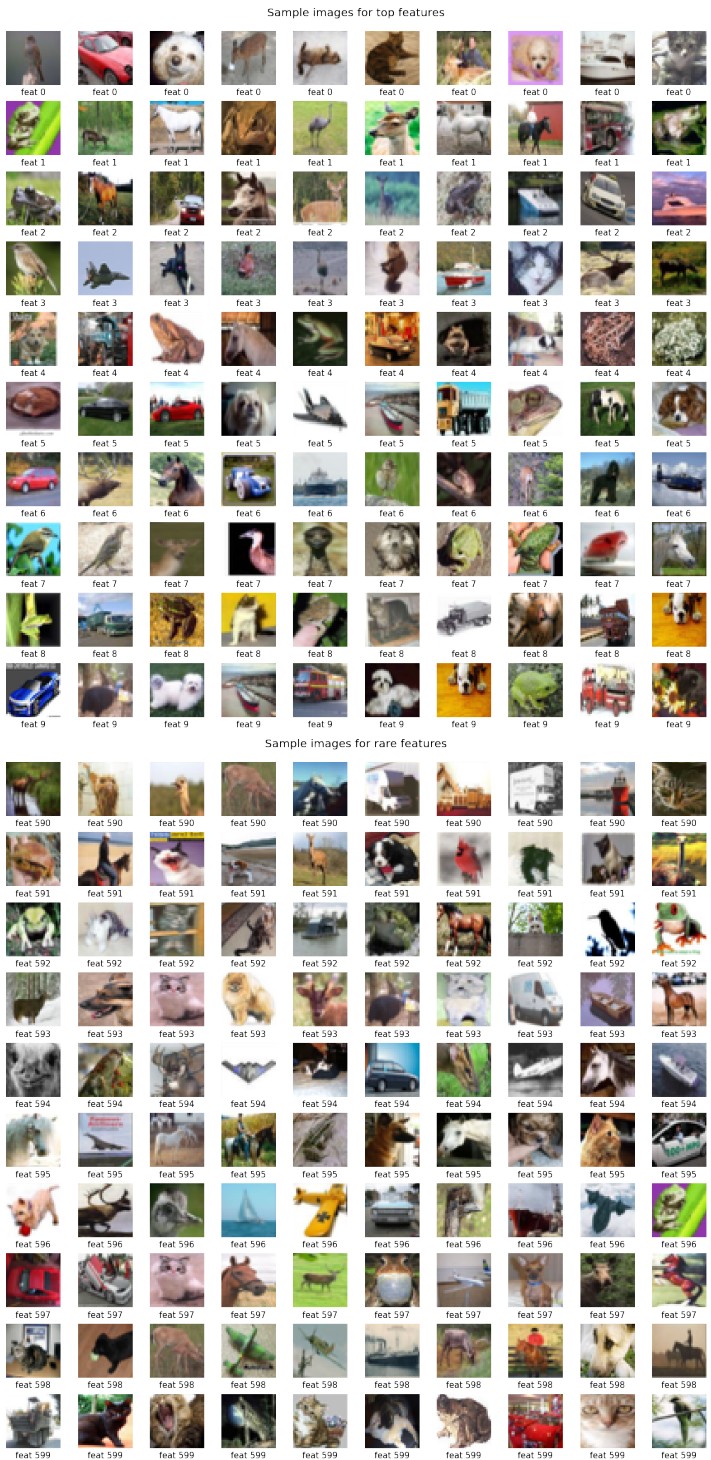

Figure 10: Random data points containing individual features. The top figure shows the sample images for dominant features and the bottom figure shows sample images for rare features. Neither shows obvious patterns, although images with dominant features do seem to be visually less complex.

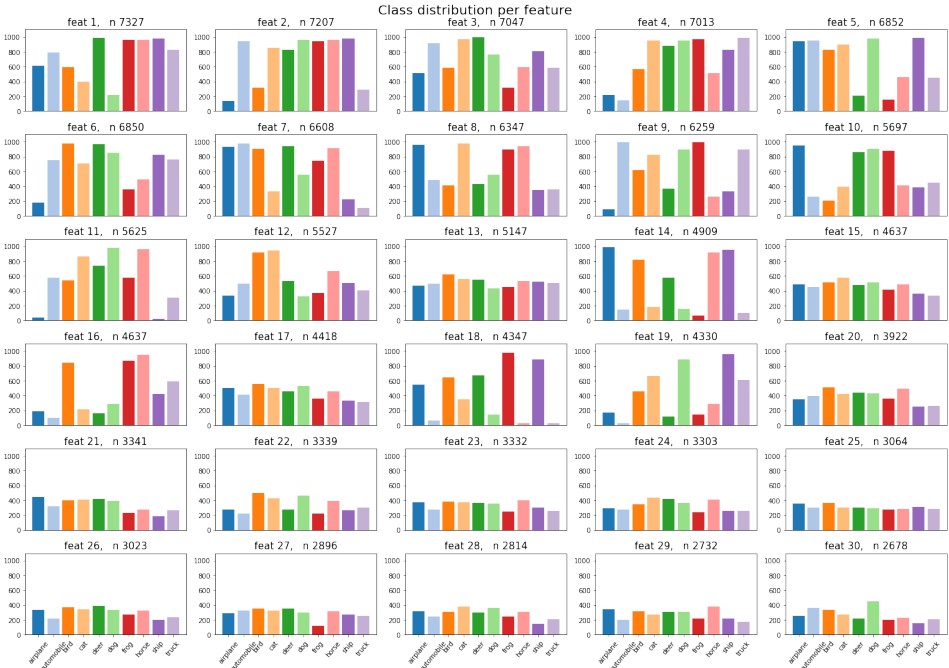

Figure 11: Frequency of top feature in each class. `n` is the feature's total number of occurrences.

## F.4 CIFAR-10 10K SUBSET EXPERIMENTS

For `10k` models, we see that the observations are largely consistent with the observations of `45k`. It is worth noting that in Figure 13e, the shared errors are generally smaller than Figure 3c and the shared errors also exhibit more variance. This may be due to the fact that when the models have low performance, their agreement behaves more randomly rather than how the agreement of `45k` behaves. We will see in Appendix F.6 that when the collection of models have different architecture, an even strong correlation between number of shared features and amount of shared error is observed.

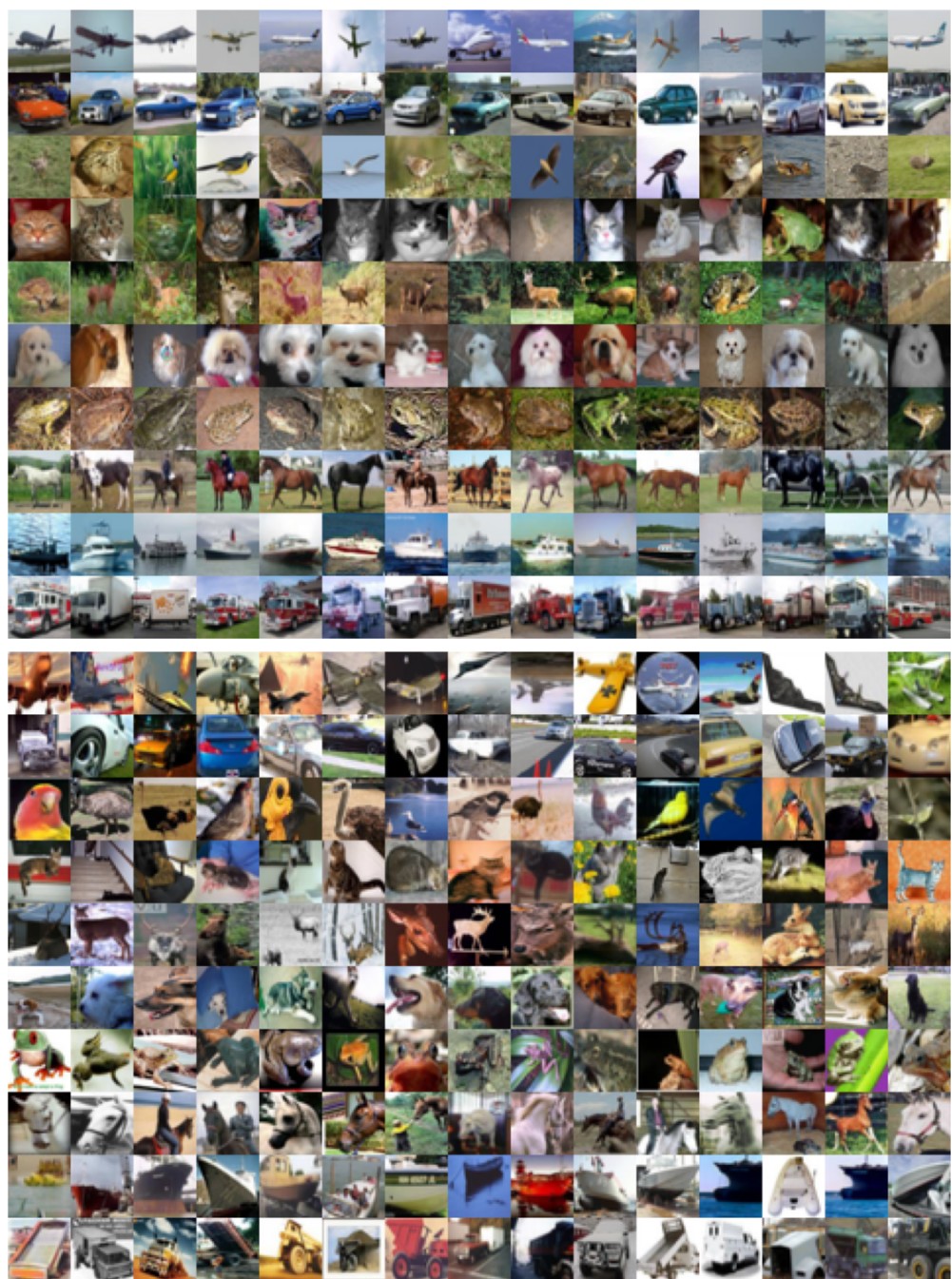

Figure 12: Visualization of images with *the least features* (top) and *the most features* (bottom) for each class of CIFAR 10 under our feature definition (defined in Section 3.1). Each row corresponds to one class of CIFAR 10 (zoom in for better viewing quality). .

## F.5 SVHN Experiments

For models trained on SVHN, we observe similar phenomena from the other experimental settings on CIFAR-10. Some notable differences include that there are much fewer low-confidence data points compared to CIFAR-10 likely because the performance of ResNet18 is higher on SVHN and the models classify most test points correctly. The absolute occurrences of different features are higher because SVHN has more test data than CIFAR-10.

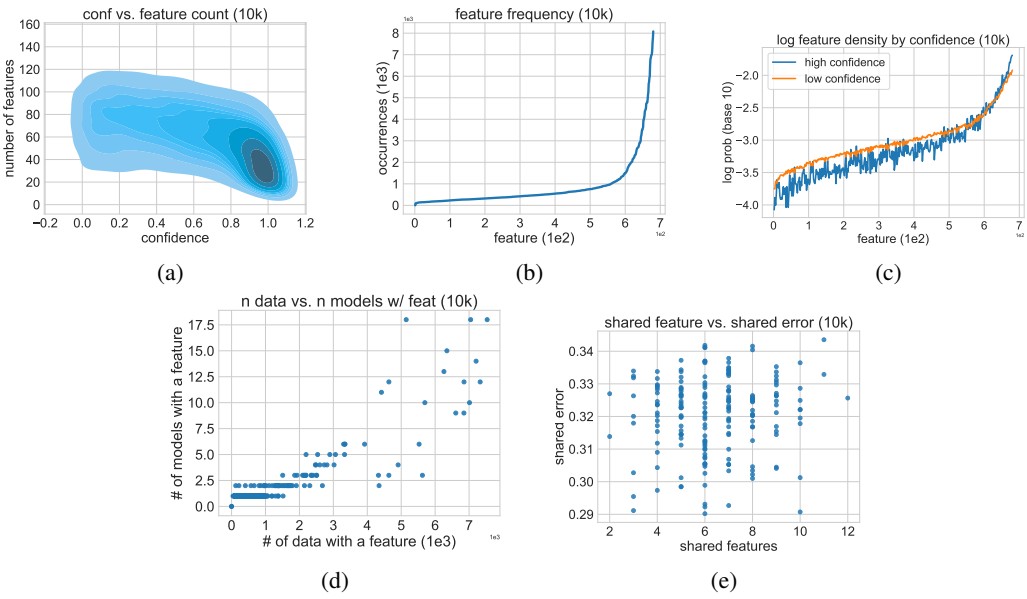

Figure 13: Same set of plots for `10k` models.

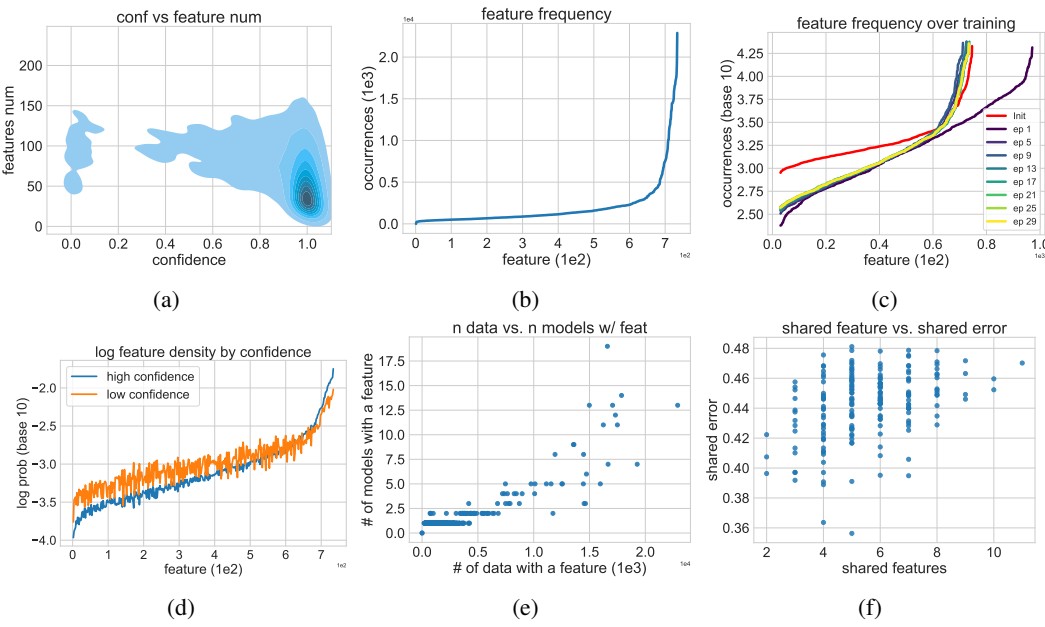

Figure 14: Same set of plots for `SVHN` models.

## F.6 Shared Features and Shared Error for Different Architectures

We see earlier that when the architectures are the same, the models naturally tend to agree with each other more. For both `45k` and `10k`, even the smallest shared error is much greater than chance. Here we will use a wide range of different architectures trained of CIFAR 10 to test the validty of our hypothesis that more shared features lead to more shared error. Figure 15 shows the number of shared features plotted against the shared error between each pair of models. Similar to Figure 3c, the shared error is almost monotonically increasing as a function of the number of shared features. If two models share a large number of features, they would tend to share high proportion of errors. If two models share a moderate number of features (4 to 11), the distribution of shared error once again appears random. However, unlike the case of same architecture, when two architectures share a

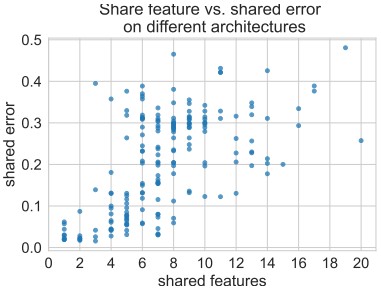

Figure 15: Shared feature against shared error for models with **different** architectures.

small number of features (1 to 3), their shared errors tend to concentrate at much smaller values. This observation indicates that while having high number of shared features generally leads to models making similar mistakes, having low number of shared features does not mean two models will have low number of shared error. Rather, different sets of features *can* still make similar mistakes. The architectures we test include:

- PreActResNet18 (He et al., 2016b)
- PreActResNet34 (He et al., 2016b)
- PreActResNet50 (He et al., 2016b)
- VGG11 (Simonyan and Zisserman, 2014)
- VGG13 (Simonyan and Zisserman, 2014)
- VGG16 (Simonyan and Zisserman, 2014)
- RegNet X200 (Xu et al., 2022)
- RegNet X400 (Xu et al., 2022)
- ResNet34 (He et al., 2016a)
- ResNet50 (He et al., 2016a)
- ResNet101 (He et al., 2016a)
- ResNeXt29 (Xie et al., 2017)

- DenseNet121 (Huang et al., 2017)
- DenseNet169 (Huang et al., 2017)
- ShuffleNetV2 with scale factor 1 (Zhang et al., 2018)
- ShuffleNetV2 with scale factor 1.5 (Zhang et al., 2018)
- ShuffleNetV2 with scale factor 0.5 (Zhang et al., 2018)
- ShuffleNetG2 (Zhang et al., 2018)
- SENet18 (Hu et al., 2020)
- SqueezeNet (Iandola et al., 2016)
- EfficientNetB0 (Tan and Le, 2019)
- PNASNetA (Liu et al., 2018)
- PNSNetA large (Liu et al., 2018)
- MobileNet V2 (Howard et al., 2017)

All models we use are from `testbed` created by (Miller et al., 2021).

### F.7 DIFFERENT CHOICES OF $\zeta$

In this section, we investigate the effect of agreement function $\zeta$ on GDE. Instead of plotting accuracy and agreement separately, we show the difference between accuracy and agreement:

$$\text{Diff} = \text{Acc} - \text{Arg}$$

The closer the difference is to $0$, the closer the system is to satisfying GDE exactly. We test three types of agreement functions, each with an adjustable parameter:

1. `constant`: This agreement function assumes that if two models share one or more features, then they have a constant probability $\eta \in [0.5, 1]$ of agreeing.

$$\zeta_{\text{const}}(k, c, \eta) = \eta$$

In the main text, we use $\eta = 0.9$.

2. `proportional`: This agreement function assumes that the probability of agreement is directly proportional to how many features two models share relative to the full model capacity. The constant of proportionality is $\eta \in (0, \infty)$ and the probability is clipped to $1$.

$$\zeta_{\text{prop}}(k, c, \eta) = \min\left(\eta \frac{k}{c}, 1\right)$$

3. `step`: This agreement function assumes that there is a threshold $\eta \in \mathbb{N}$. If the number of shared features is above $\eta$, then the probability of agreement is $1$. Otherwise, the probability of agreement is some constant $\theta \in [0.5, 1]$.

$$\zeta_{\text{step}}(k, c, \eta, \theta) = \theta \cdot \mathbb{1}\{c \leq \eta\} + \mathbb{1}\{c > \eta\}$$

For these simulation, we use $\theta = 0.8$ since $\eta$ has a much greater effect on GDE.

We vary the values of $\eta$ for each agreement function and show the results for different values $p_d$, $c$, coupled $n_r, n_d$ and coupled $t_r, t_d$ in Figure 16. Each row corresponds to a different agreement function and from top to bottom are `constant`, `proportional`, and `step`.

For `constant` (top row), $\eta$ ranges from $0.5$ to $0.95$. We see that for $c, p_d, n_r \propto n_d$, the range of variation in difference is consistently small when $p_d$ is sufficiently large. $t_r \propto t_d$ deviates from this behavior where different $\eta$'s behave more differently as $p_d$ increases. For this scenario, we see that larger $\eta$ are closer to GDE.

For `proportional`, $\eta$ ranges from $1$ to $2.8$. We see that the difference is generally large for all values of the parameters. Suggesting that $\zeta_{\text{prop}}$ may not be a good approximation for how models agree in practice.

For `step`, $\eta$ is an integer that ranges from $0$ to $9$. We see that the differences are more robust to different values of $\eta$ than the other agreement functions. This suggests $\zeta_{\text{step}}$ could be a good approximation to how models agree in practice. This observation is consistent with Figure 3c, Figure 13e, and Figure 15 — when models do not share many features, the shared error (therefore, agreement) is spread out but have similar expected values; when models share a large number of features, the probability of agreement increases significantly.

An important observation from these simulations is the importance of $t_d$ and $t_r$. These quantities can be interpreted as proxies for the complexities of the entire dataset. $t_r$ in particular represents the patterns in the data that are rare. The larger $t_r$ is, the more *diverse* or *noisy* the dataset is. According to our framework, this quantity can have large impact on the behaviors of accuracy and agreement.

Another important observation is that $p_d$ needs to be sufficiently large for GDE to hold strongly. This is roughly equivalent to requiring the feature distribution to be long-tailed, which is true in practice.

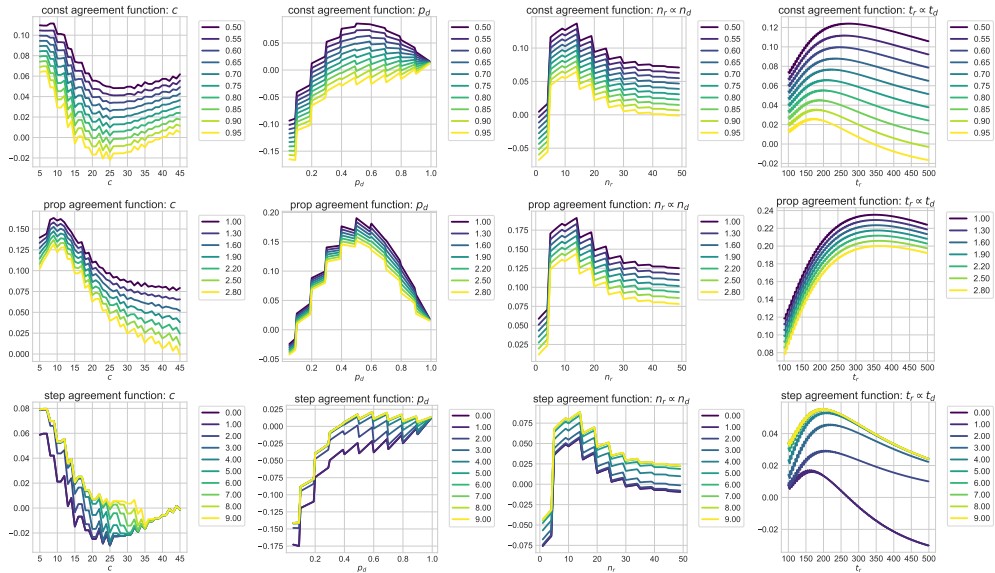

Figure 16: Difference between the analytical agreement and accuracy for different choices of agreement functions $\zeta$ and different values of parameters. From top to bottom are `constant`, `proportional`, and `step`.

## F.8 MERGING CLASSES

For this experiment, we merge different classes of CIFAR 10 to form superclasses. Since the individual images are not modified, we expect the majority of features that identify individual classes to also identify the superclasses well (although there may be interferences between features of different classes). Thus, we would expect the ratio between features stay approximately constant.

The specific superclasses are:

- 2 superclass: {airplane, automobile, bird, cat, deer}, {dog, frog, horse, ship, truck}
- 3 superclass: {airplane, automobile, bird}, {cat, deer, dog}, {frog, horse, ship, truck}
- 5 superclass: {airplane, automobile}, {bird, cat}, {deer, dog}, {frog, horse}, {ship, truck}

Finally, the 10 classes case corresponds to the regular CIFAR 10 classification. The experiments are repeated for 6 random seeds.

## F.9 PARTITIONING DATA

For this experiment, we partition CIFAR based on the intensity of blue pixels. More concretely, let $\boldsymbol{x} \in [0, 1]^{32 \times 32 \times 3}$ be an image where the last channel is the RGB value of the pixel. We compute its *blue intensity*, $b$, as:

$$b(\boldsymbol{x}) = \frac{\sum_{i=1}^{32} \sum_{j=1}^{32} \boldsymbol{x}_{i,j,2}}{\sum_{i=1}^{32} \sum_{j=1}^{32} \sum_{k=1}^{3} \boldsymbol{x}_{i,j,k}}. \tag{49}$$

Intuitively, this value captures how much the blue channel "weighs" in the whole image. We compute this value for all $\boldsymbol{x}_i$ and compute the CDF, $F$, of $b(\boldsymbol{x}_i)$ over the training dataset $\mathbf{X}_{\text{train}} = \{\boldsymbol{x}_0, \boldsymbol{x}_1, \dots, \boldsymbol{x}_{50000}\}$. Then we partition the data into groups:

$$G_i = \{\boldsymbol{x} \in \mathbf{X} \mid 0.2i < F(b(\boldsymbol{x})) \leq 0.2(i+1)\}, \tag{50}$$

for $i \in \{0, 1, 2, 3, 4\}$. For the test data, we partition according to the CDF of the **training data**, i.e., training and test use the same threshold. In Figure 17, we show random samples of images from each partition based on the blue intensity. We can see that in group 0, the examples seem more visually complex and diverse, which could lead to a larger number of rare features (i.e., larger $t_r$, the total number of rare features). The experiments are repeated for 6 random seeds.

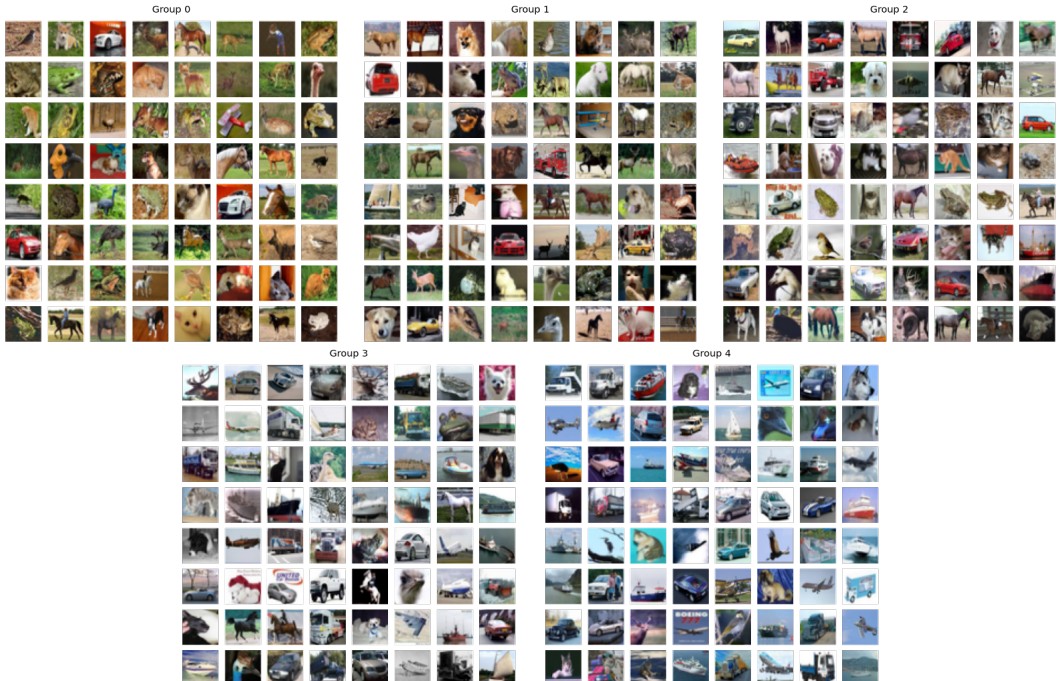

Figure 17: Visualization based on data partitioning based on blue intensity.

## F.10 EMPIRICAL OBSERVATIONS UNDER DIFFERENT HYPERPARAMTERS

The construction of the interaction tensor involves several hyperparameters. In this section, we study how sensitive the empirical observations are to the changes in hyperparameters. Specifically, we study the effect of $\gamma_{\text{corr}}$ which decides whether two principal components of two models are capturing the same feature, and $k$, the number of principal components used for each model. We show the results for $\gamma_{\text{corr}} = 0.5$ and $\gamma_{\text{corr}} = 0.7$ in Figure 18 and Figure 19, and the results for $k = 30$ in Figure 20.

We observe that the most significant impact of $\gamma_{\text{corr}}$ is on the total number of features because having a higher threshold would mean that more features would be counted as different features if they do not have highly correlated responses to data; this would increase the total number of features. Interestingly, the qualitative observations remain the same which means the conclusion is robust to the choice of these hyperparamters.

Similarly, too many principal components would inflate the number of rare features (features at the tail). Empirically, this would affect Observation 1 by inflating the length of the tail (but it is still long-tailed), but it should not affect Observation 2-4. For 2, the high-confidence data would still have a few dominant features. For Observation 3, since these inflated rare features do not usually appear in more than one model and are not useful for classification, this observation is not changed either. For Observation 4, following the previous reasoning, models with similar features would still make similar mistakes.

Note that in our experiments, 30 is the minimum number of principal components required to retain the full accuracy on CIFAR10 although this can change depending on the dataset (note that the plots are almost identical to Figure 2 and Figure 3). Since more principal components only affect the rare features, the effect of moderate overcounting is not significant. For the main experiments, we chose 50 principal components that are sufficient for retaining the full classification accuracy on the test data but not too large.

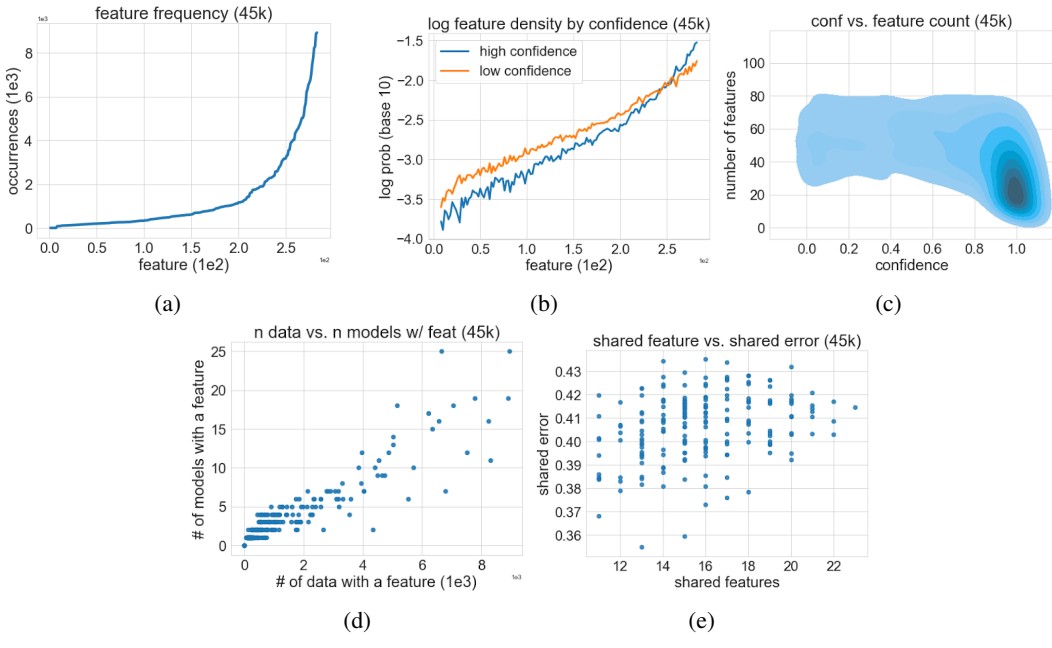

Figure 18: $\gamma_{\text{corr}} = 0.5$

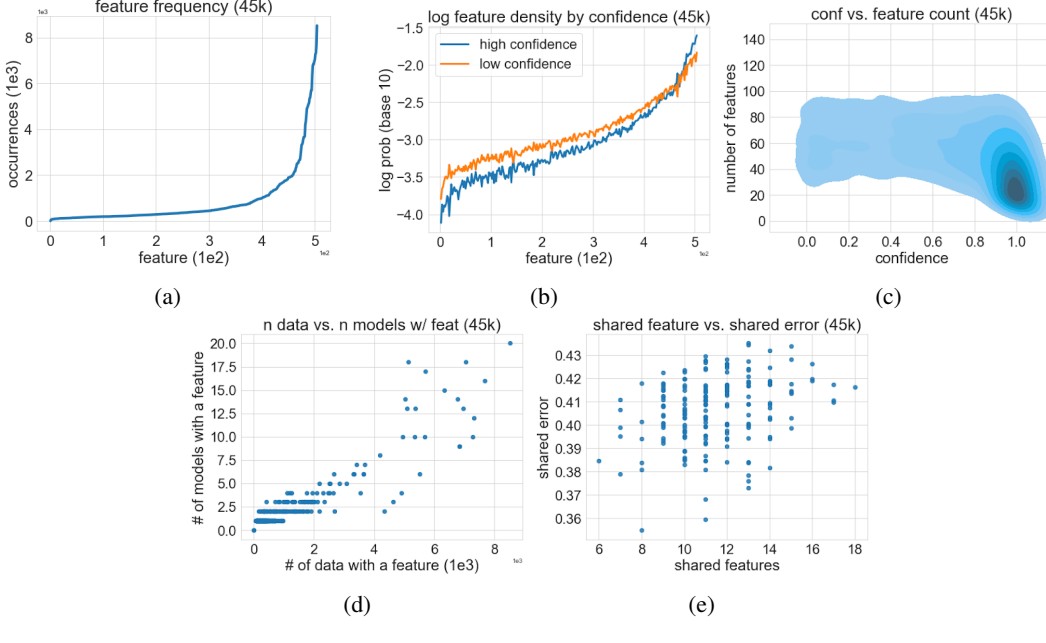

Figure 19: $\gamma_{\text{corr}} = 0.7$

## F.11 PROGRESSION OF FEATURES AT DIFFERENT DEPTH DEPTH

In the main text, we studied the distribution of features at the input of the final linear of the model. As we mentioned earlier, this choice is to avoid *superposition* (Elhage et al., 2022) where a model leverages the non-linearity to model multiple features using a few features. This phenomena cannot happen at the final layer since only the linear component of the representation can be used to make decision. In general, we want to emphasize that features are not always well-defined object. In all but the simplest cases can we say what the feature of an input is concretely.

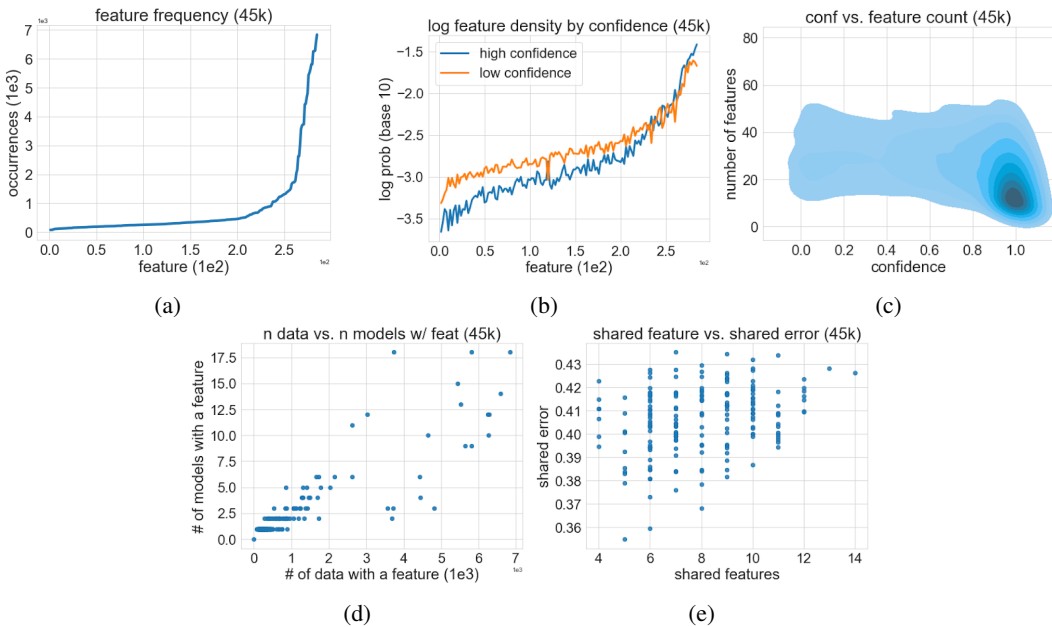

Figure 20: `45k` using 30 principal components per model.

In fact, it is not particularly productive to talk about features without talking about the model used to classify the data. One may be interested in resorting to human interpretation but the human cognition is in itself a classifier with features, and a machine learning model may or may not have the inductive bias to automatically learn the same features. For example, humans may think the face of a puppy is the defining feature of a puppy, but a neural network may consider the fur or paws as the most important features.

In a deep neural network, we often say that the model learns *hierarchical features*. These features are combined to form more complicated features (Olah et al., 2017). However, this also means that not all information present in these features is necessary for performing classification. From this perspective, the final layer features may be viewed as the "greatest lower bound" of this feature hierarchy as the model can no longer compress the feature beyond the final layer and whatever features present at this layer is the feature used to predict the classes. As we can see in Figure 10 and Figure 11, these features do not correspond to human perception in an obvious way and are thus not highly interpretable.

That being said, looking at the feature distribution at different depths of the model could reveal interesting observations about the evolution of features within the model at a macro level. The ResNet18 architecture we used can be broken down into 4 residual blocks. We refer to the features at the output of each block as `blockn` where $n$ is the index of the layer (e.g., the results in the main paper would be `block4`). In Figure 21, 22 and 23, we present the feature distribution and associated artifacts for `block1`, `block2` and `block3`. We observe that $O.1$, $O.3$ and $O.4$ generally still hold across different depth. However, the separation in data with different confidence ($O.2$) tends to happen at deeper layers. In the earlier layers, the features of data at different confidence levels are not distinguishable, but this separation becomes more pronounced at deeper layers (subplot (b) and (c)). In other words, the representations for data with high confidence become increasingly similar and compressed the deeper they are into the model. The rare data that are less compressible are encoded with additional dedicated features (rare features) that are not used for the dominant data.

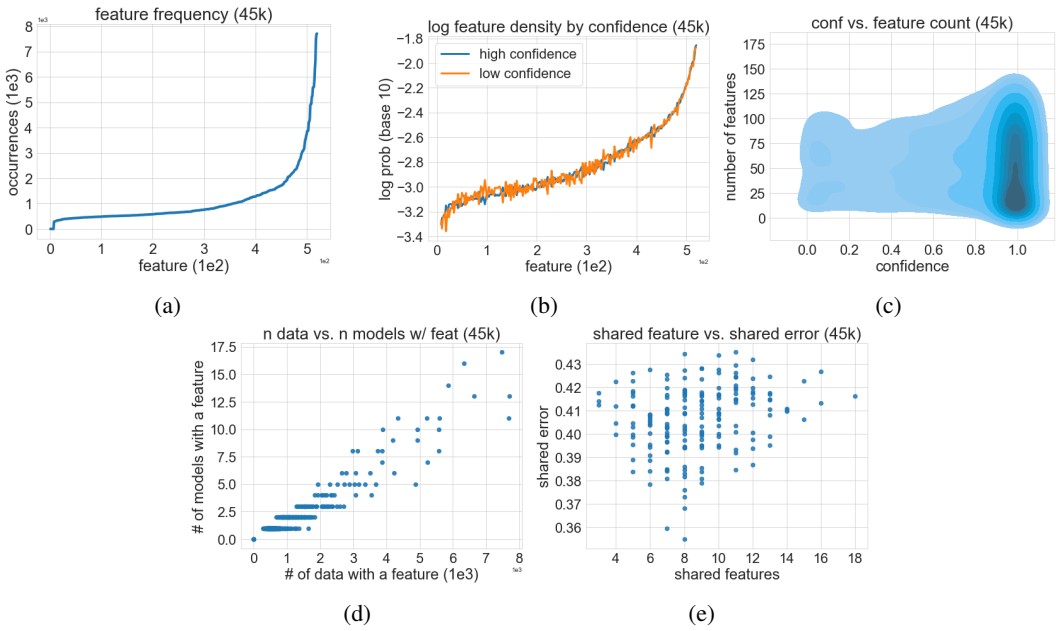

Figure 21: Features distribution of `45k` at `depth1`.

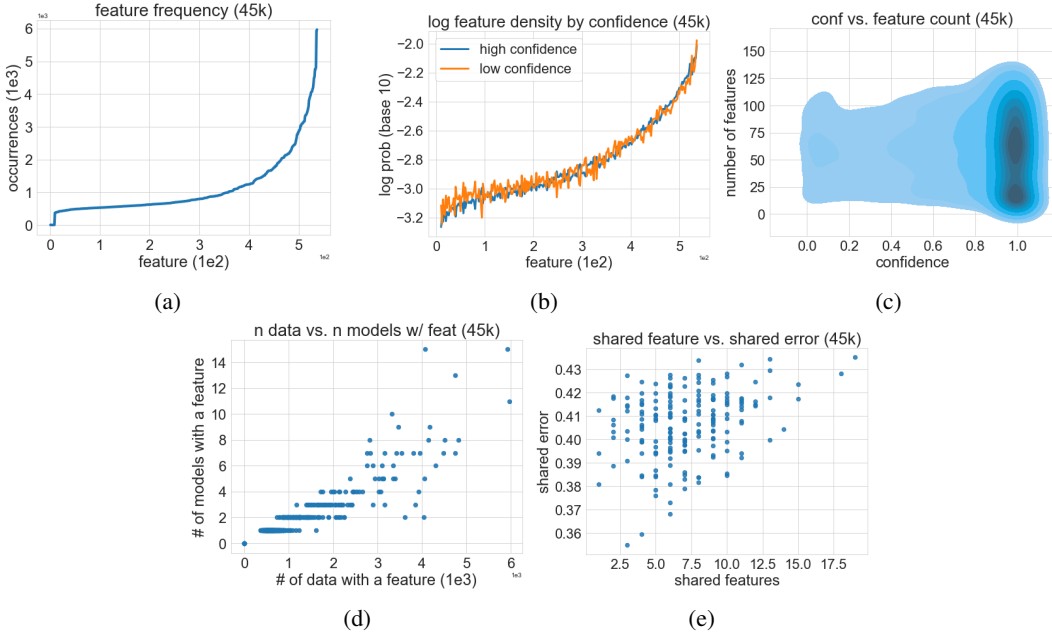

Figure 22: Features distribution of `45k` at `depth2`.

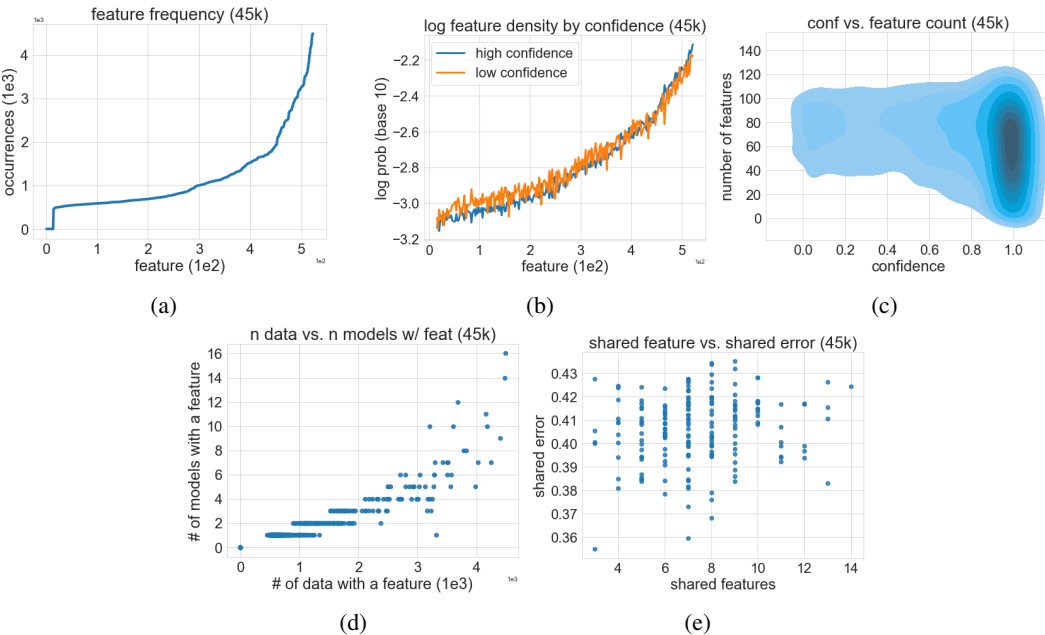

Figure 23: Features distribution of `45k` at `depth3`.

