# OpenReview forum: "On the Joint Interaction of Models, Data, and Features"
_ICLR.cc/2024/Conference — ICLR 2024 oral_

### Official Review · Reviewer_PinD · 2023-10-25

**Soundness:** 3 good
**Presentation:** 4 excellent
**Contribution:** 4 excellent
**Rating:** 8
**Confidence:** 4

**Summary:**

The authors propose a novel framework for analyzing feature learning in deep NNs, from the perspective of observed phenomena occurring in trained networks. The learnt features in a trained model are defined as the principal components of the last layer's activations, and the proposed interaction tensor examines these feature representations of a dataset for a set of models. The interaction tensor is used to empirically validate several common-sense intuitions, such as feature distribution being long-tailed, which then leads to a further abstracted framework for combinatorial-style analysis. This framework is able to theoretically demonstrate empirical phenomena from the observations as well as prior works such as GDE.

**Strengths:**

- Due to the increasingly complex nature of deep networks, the concept of studying feature learning's observed phenomena like a natural science is a useful supplement to more derivation-based frameworks such as NTK
- The combinatoric analysis is a creative and original perspective for studying feature learning. While the simplifications might seem a little overly-simplistic at first, the experiments in section 6 and observations in section 4 serve as adequate motivation
- The resulting findings of the framework not only validate but add nuance to understanding of prior phenomena such as GDE

**Weaknesses:**

- Like any theoretical framework, many assumptions went into the combinatorial analysis. In particular, this framework assumes features and datapoints are each either dominant *or* rare, a dominant/rare datapoint always has the same number of dominant/rare features, etc
- The framework is less useful for understanding the learning process of networks, such as why some runs might collapse while others successfully learn the desired features
- Some minor typos: section 5, data generating process paragraph's second-to-last sentence samples $n_r$ rare, not dominant, features. Appendix C, equation line 34 the two $\not = \emptyset $ could maybe instead be $= \emptyset$

**Questions:**

- While simplifying features and datapoints to be either dominant *or* rare is good enough for the section 6 experiments, have you considered modeling the rare-dominant variation as a spectrum instead of a binary? For instance, looking at figure 2b, I'm not sure where I would want to draw a line to separate the rare from the dominant features. Even putting the line somewhere around x=6.5, there is still a relatively large deal of variation of frequency of occurrence in both the rare and the dominant feature types

---

> ### Author Response · Authors · 2023-11-16
> **Response to Reviewer PinD**
>
> Thank you for your thoughtful feedback! We are glad that you still find our work original and creative.
>
> > many assumptions went into the combinatorial analysis.
>
> We fully agree that many assumptions were made. The way we see it is that deep learning's complexity necessitates some level of simplification for tractable analysis. In this case, since the resulting framework is still useful for understanding empirical phenomena, we feel the assumptions are justified.
>
> > The framework is less useful for understanding the learning process of networks…
>
> This is correct. The framework as it stands does not describe the learning process, which is an extremely complicated subject itself. However, we postulate that techniques similar to [1] could be used to study the optimization of a feature model like ours in the future.
>
> > typos
>
> Thank you for the close reading and for catching these typos! We have fixed them in the revision.
>
> > have you considered modeling the rare-dominant variation as a spectrum instead of a binary?
>
> This is an excellent question! We have indeed considered the possibility of modeling it as a spectrum, which we discussed in Appendix D.1 and D.4. In our preliminary investigation, we found that modeling the feature spectrum may require introducing more free parameters (e.g., Zipf’s distribution), and can drastically increase the complexity of analysis. Given that this is our first foray into the topic, we opted for the simpler binary model. This is an important problem and we plan to explore it in more detail in future works.
>
> We hope these responses have addressed some of the questions and we are open to further discussion. Thank you again for your constructive feedback!
>
> **Reference**
>
> [1] Towards understanding ensemble, knowledge distillation and self-distillation in deep learning. Allen-Zhu et al.

---

> > ### Comment · Reviewer_PinD · 2023-11-19
> >
> > Thank you for the detailed response; I agree with the justifications provided and also appreciate your efforts with the other reviewer to improve the notation. I think the work presents a very novel analysis perspective and will raise my review to an 8.

---

### Official Review · Reviewer_MLur · 2023-10-30

**Soundness:** 3 good
**Presentation:** 4 excellent
**Contribution:** 3 good
**Rating:** 6
**Confidence:** 4

**Summary:**

This paper introduces the interaction tensor, for empirically analyzing the interaction between data and model through features. Based on some observations using this tensor, they propose a very simple toy model (a combinatorial model) for feature learning. They show that this model also exhibits Generalization Disagreement Equality (GDE). Finally, the authors use their model to provide data distributions that break GDE in real world experiments.

**Strengths:**

- The problem of feature learning is quite important and there has been a lot of attempts for gaining a better theoretical understanding of it in recent years.

- The authors are able to come up with a toy model that shows GDE which is a very important phenomenon (and it does need the explicit assumption of calibration).

- The model that the authors propose is simple and it can be analyzed fully.

- The paper is very well-written.

**Weaknesses:**

- I definitely agree with "the empirical phenomena of deep learning can be understood at many different layers of
abstraction". However, I think the model proposed in this paper is too simplistic. The implicit biases of deep learning are core to some of the merging phenomenon that we see these days and the models that the authors propose fails to capture that. I also think that a good toy model should leave the door open to generalizations and getting closer to real world practice (for example, for the random features model of deep learning, there is a very obvious way to move towards making it more realistic). But the models that the authors propose is too abstract and it is not clear what simplifications are made to the real problem to arrive at the proposed model.

- Although the model is based on some observations using the interaction tensor, I still find the model to be not very well motivated. Any insights on how this can relate to the training of deep nets?

- It is not very clear how the authors set the hyperparameters in their model (e.g., the thresholds).

- The notations are a bit confusing (i, j, k, etc.). I suggest authors avoid using these generic letters. Also, the paper will benefit greatly from a figure that summarizes all the notations (p_d, p_r, etc.). It will also help explain the method.

- The authors "prove" GDE in their model without the explicit assumption of calibration. But the model is very abstract/high-level and I'm not sure if the assumptions that they make are stronger or weaker than calibration.

- The theoretical understanding of feature learning is not as rudimentary as the authors claim. For example,

[1] Alex Damian, Jason Lee, and Mahdi Soltanolkotabi. Neural networks can learn representations with gradient descent, 2022.

[2] Zhichao Wang, Andrew Engel, Anand Sarwate, Ioana Dumitriu, and Tony Chiang. Spectral evolution and invariance in linear-width neural networks, 2022.

[3] Eshaan Nichani, Alex Damian, and Jason D Lee. Provable guarantees for nonlinear feature learning in three-layer neural networks, 2023.


[4] Yatin Dandi, Florent Krzakala, Bruno Loureiro, Luca Pesce, and Ludovic Stephan. Learning two-layer neural networks, one (giant) step at a time, 2023.

[5] Behrad Moniri, Donghwan Lee, Hamed Hassani, and Edgar Dobriban, A theory of non-linear feature learning with one gradient step in two-layer neural networks, 2023.

and many more.

**Questions:**

- Looking at figure 3 (a) and 3 (c), it seems that the observations made from them are not that significant. Am I missing something? How does the choice of the thresholds affect your observations?

---

> ### Author Response · Authors · 2023-11-16
> **Response to Reviewer MLur (1/2)**
>
> Thank you for your detailed review and thoughtful feedback. We are glad that you find the problem we study timely and important. Below are our responses to your concerns:
>
> > I definitely agree with "the empirical phenomena of deep learning can be understood at many different layers of abstraction". However, I think the model proposed in this paper is too simplistic…
>
> We definitely agree that our model is unconventional and more abstract than most existing works but similar assumptions have been made in prior work [6]. As is the case for all theoretical models, simplifications are required to keep analysis tractable. Despite the simplification, our model already exhibits some interesting behaviors and offers insight into empirical phenomena so we believe that it adds value to the current discussion on feature learning.
>
> In the case of random feature models, we feel that it is a little unfair to say in hindsight that there was a very obvious way to move towards making it more realistic. When NTK first came out in 2018, it was thought to be very promising. Indeed, despite its simplification, it offers clear value in the analysis of some phenomena in deep networks. However, it only later became clear that there was a big gap between NTK and real deep learning (despite many efforts to close this gap). More recent works (many of which you already brought up) rely on quite distinct analysis tools than those used in NTK and, still, it’s not clear if any of them is close to real-world practice.  As such, while we agree (and have acknowledged in Section 5, Appendix D.1 and D.4) that there is a gap between our proposed framework and practice, we feel it’s a tall order for the current state of deep learning theory to have no strong assumptions.  Just like NTK analysis, we hope that our approach will ultimately prove valuable (albeit along different dimensions), despite its simplicity.
>
> > Although the model is based on some observations using the interaction tensor, I still find the model to be not very well motivated. Any insights on how this can relate to the training of deep nets?
>
> The primary goal of our work is to investigate the effect of different features being learned by different models on GDE and the calibration of deep ensembles, both of which are robust empirical observations. While our model does not directly provide insights into training, understanding these feature interactions may indirectly inform training strategies in the future. For example, [6] shows potential avenues for how observations like ours may transfer to training.
>
> > It is not very clear how the authors set the hyperparameters in their model (e.g., the thresholds).
>
> As detailed in Appendix F.10 (referred to at the end of Section 4), our conclusions are robust across a range of parameter settings so we believe that the settings used in our paper should be appropriate for most situations.
>
> > The notations are a bit confusing ($i, j, k$, etc.). I suggest authors avoid using these generic letters. Also, the paper will benefit greatly from a figure that summarizes all the notations ($p_d, p_r$, etc.). It will also help explain the method.
>
> We apologize for the confusion. There is already a centralized list of notations in Appendix A which we could not include in the main text due to space constraints (this is referred to in the second paragraph of section 5). Beyond this, could you be more specific about where the $i,j,k$ are causing confusion so we could better address the confusion? $i,j,k$ are used to denote an arbitrary member of the ordered collection so we feel this is in accordance with the standard usage.
>
> > But the model is very abstract/high-level and I'm not sure if the assumptions that they make are stronger or weaker than calibration.
>
> We want to clarify that we *do not* claim that the assumptions we make are weaker than calibration (since neither is a clear superset of the other), but we do believe that it sheds more light on the underlying reasons behind GDE. Only stating the model is calibrated does not tell us much about the model or the data, but our assumptions do. In fact, our framework tells us how one can break our assumption (and calibration) through data distribution interventions. Of course, it is an important open question why so many natural distributions seem to satisfy these assumptions.

---

> ### Author Response · Authors · 2023-11-16
> **Response to Reviewer MLur (2/2)**
>
> > The theoretical understanding of feature learning is not as rudimentary as the authors claim
>
> We sincerely apologize for the characterization. Perhaps “rudimentary” is not the best description. We have revised the phrase to be “still in early development” to better convey the sentiment. However, we do feel that it is reasonable to say that we still have much to understand about feature learning.
>
> To briefly address the specific works you mentioned: these models generally assume the input data are isotropic Gaussian and the function class is a 2 layer MLP. The feature is usually a linear function of the input and often special training algorithms are needed (e.g., layer-wise training). A notable exception is [3] which learns non-linear features but still requires layer-wise training.
> Most of these works are quite recent (and all came out after we started this project): [3,4,5] showed up in October 2023 and [2] was updated in November 2023. In general, it is hard to say how well they encapsulate the full complexity of deep learning, or how feasible it is to apply them to modern architectures and datasets. Nonetheless, the goal of this work is not to compare existing theoretical works on the topic or discuss their merits and shortcomings, but rather to identify what realistic (perhaps strong) modeling assumptions to make when studying feature learning. Given the recently increasing interest in the subject, we view these works and our approach as complementary components of a broader exploration into feature learning. We have added a discussion of these works to Appendix D.1, but would be happy to revise them if you have further suggestions.
>
> We hope these responses address your concerns and we are open to further discussion. Thank you again for your constructive feedback!
>
> **Reference**
>
> [6] Towards understanding ensemble, knowledge distillation and self-distillation in deep learning. Allen-Zhu et al.

---

> ### Comment · Reviewer_MLur · 2023-11-17
>
> I thank the authors for writing a detailed response to my comments and updating the manuscript. I have read all the reviews, comments, and the revised manuscript. I agree with the authors that this submission does indeed shed some light on the reasons behind GDE.
>
> However, I still have the following comments regarding the submission:
>
> 0- I think the authors forgot to address this point: "Looking at figure 3 (a) and 3 (c), it seems that the observations made from them are not that significant. Am I missing something? How does the choice of the thresholds affect your observations?". Can the authors please clarify on this?
>
> 1- When NTK was first introduced, it was obvious that it is a first order Taylor series expansion around initialization (that is accurate in the infinite width regime) and it was obvious that adding more terms will make it more realistic (but maybe harder -- or much harder -- to analyze).
>
> Also, when people study shallow neural networks (or infinitely deep neural networks), the assumptions that they make is explainable. We know what aspects they are missing out on. We know that they need to add more layers, we know the training procedure is very specialized, etc. However, this is not the case for your model. I am not sure what your assumptions/simplifications are to the real problem.
>
> This is a limitation of the current work. I think a "limitations" section discussing these issues might benefit the paper.
>
> 2- The notation of this paper is not well optimized, and it is making the paper very hard to read at some points. I know the authors are also aware of this (e.g., the footnotes in page 4). Adding a table discussing notations in main text (I know there is a page about this in appendix) will greatly benefit the paper -- at least if (when) the authors plan to publish an extended manuscript on a preprint server later.

---

> > ### Author Response · Authors · 2023-11-18
> >
> > Thank you for the continued engagement and dedication to the review process! We address your remaining concerns as follows:
> >
> > > 0
> >
> > Apologies for missing the first part of your question earlier. In 3 (a), the observation we are relying on is that data points with a small number of features are much more likely to have very high confidence under the ensemble and vice versa. For a concrete example, it is much more likely for a data point with 0.8 confidence to have 25 features than a data point with 0.4 confidence to have 25 features. As such, we believe that it’s fair to say that data with fewer features tend to have higher confidence. For 3(c), we claim that the more features the models share, the lower bound on their shared errors becomes higher, which can be clearly observed in the figure. This *does not* mean that the fewer features the models share, their shared errors would necessarily be smaller. Instead, both models would make random guesses (recall that we make similar assumptions in the conceptual model). This means that a pair of models can have a high or low amount of shared errors due to randomness, but the lower bound on the shared error will become higher if they share more features. We believe that the figure supports this claim.
> >
> > > 1
> >
> > Thank you for clarifying what you meant about the limitations. We agree with your assessment of NTK and shallow neural networks, and we absolutely agree that we should be transparent with the limitations of the framework. Appendix D.3 (previously D.4) is meant to discuss the limitations of the model but we see that it may not be as clear as we hoped. As such, we have expanded Appendix D.3 to be more explicit about these limitations. This was already referred to at the beginning of section 5 where we introduced the model and in the conclusion, but once again, due to space limits, we could not include them in the main text.
> >
> > > 2
> >
> > Thank you for clarifying this and pointing out the notation issues. We have streamlined the notations in Section 3 for better clarity so the footnotes are no longer needed.
> >
> > Specifically, we have changed to use $C$ for the number of classes instead of $K$ and use $K$ to denote the number of top principal components, so all capital letters refer to a fixed number (e.g., $N$ for the total number of data, $M$ for the number of models, etc). Lower case of $N,M,K,T$ will refer to a particular index for what the object the capital letters refer to, so $k$ will always refer to the feature, $m$ will always refer to the model and $n$ will always refer to data, etc.
> >
> > One exception is the k-partite graph. Since this is the standard reference to the problem we use $\mathtt{k}$ to differentiate it.
> > Another one is that $i,j, a, b$ are used instead of $m$ and $k$ to refer to a pair of models and features around equation 2.
> > We believe that this should resolve the ambiguity but if there are any concerns regarding the notation left please let us know.
> >
> > Regarding the table of notation, we fully agree that a table would be great to have. Unfortunately, so far we have not been able to find a way to squeeze it into the text. And we will definitely include it in the extended version.
> >
> > ------------
> >
> > We hope these revisions and clarifications address your concerns. Please let us know if you have any further questions.

---

> > > ### Comment · Reviewer_MLur · 2023-11-18
> > >
> > > I thank the author for their response.
> > >
> > > I am satisfied with the answer to point 0.
> > >
> > > I still think point 1 is a limitation of the current study, but the "limitations" section is now making everything much more transparent. I thank the authors for expanding it.
> > >
> > > I also thank the authors for the effort to make the notation better. I think the revision made the paper much more readable.
> > >
> > >
> > > I will raise my score from 5 to 6.

---

### Official Review · Reviewer_PxMC · 2023-10-31

**Soundness:** 3 good
**Presentation:** 3 good
**Contribution:** 3 good
**Rating:** 8
**Confidence:** 4

**Summary:**

This paper focuses on the interaction between data and model through feature to understand deep learning from the feature learning perspective. Based on their observations, they propose a framework to characterize the quality and process of feature learning, with theoretical support. Some empirical results are provided to validate their approach.
.

**Strengths:**

1) To catch on feature learning process during deep learning is a key problem in community.
2) Paper provides a practical framework with solid theoretical analysis.
3) Empirical results on different datasets are provided. And clear experimental details are listed.

**Weaknesses:**

Figure 1 is a bit vague. It is recommended to replace it with a clearer version.

**Questions:**

See weakness.

**Details Of Ethics Concerns:**

nan

---

> ### Author Response · Authors · 2023-11-16
> **Response to Reviewer PxMC**
>
> Thank you for the support of our paper! We are glad that you find our work valuable.
>
> Regarding Figure 1, to better address your concerns, could you elaborate on which aspects of the figure you found unclear? This will help us make more targeted improvements.
>
> The goal of Figure 1 is to demonstrate that our definition of features can capture images with redundant information and images that contain uncommon instances of a given class. To make this clearer, we have updated the figure in the paper with texts to better reflect the core idea. Does this resolve your concerns? If not, we’d be happy to make more adjustments.

---

> > ### Comment · Reviewer_PxMC · 2023-11-22
> >
> > Thanks for the clarifications. I will maintain my initial score.

---

### Official Review · Reviewer_AyvM · 2023-10-31

**Soundness:** 3 good
**Presentation:** 3 good
**Contribution:** 3 good
**Rating:** 8
**Confidence:** 2

**Summary:**

The paper proposes the interaction tensor which is a binary 3-dimensional tensor T describing the presence of certain features in both model and data point, i.e., if $T_{tmn} = 1$ implies that $t_{th}$ feature is present in both $m_{th}$ model and $n_{th}$ data point. The construction of the interaction tensor is based on the correlation analysis between PCA-reduced features of penultimate layers of the collection of models on the given dataset. Authors utilize this interaction tensor to empirically analyze properties of feature learning and propose a feature learning model based on their observations. Using this feature learning model they focus on analysing recently proposed Generalization Disagreement Equality and given which conditions GDE can arise.

**Strengths:**

Understanding how models learn is an important topic in modern deep learning. Authors build the new framework to describe feature learning from the different perspective which allows to describe recently observed phenomenas. I think that this new perspective provides a valuable contribution to the community and can facilitate further developments in this area. In addition, I personally liked the construction of a natural dataset on which deep ensemble is not well-calibrated in-distribution and where GDE fails.

**Weaknesses:**

Honestly, I don't see obvious weaknesses of the proposed framework and study.

**Questions:**

1. Given that construction of interaction tensor depends on thresholding ($\gamma_{corr}$ and $\gamma_{data}$), how important are these hyperparameters? How to properly set them?
2. Currently theoretical framework analyzes binary classification, does the analysis extend to multi-class classification?
3. Does the framework allow for introducing distribution shifts and etc?

---

> ### Author Response · Authors · 2023-11-16
> **Response to Reviewer AyvM**
>
> Thank you for your supportive feedback on our work. We are glad that you find our new perspective on feature learning interesting. Below are our responses to your questions:
>
> > how important are these hyperparameters? How to properly set them?
>
> In Appendix F.10, we conducted an ablation study and found that our qualitative conclusions are robust to variations in these hyperparameters. Therefore, we believe the values used in our paper should be appropriate for most applications.
>
> > Currently theoretical framework analyzes binary classification, does the analysis extend to multi-class classification?
>
> We discuss the necessary steps for the extension to multi-class classification in Appendix D.3.
>
> >Does the framework allow for introducing distribution shifts and etc?
>
> This is a great question. We believe our framework can accommodate distribution shifts. For instance, the simplest form of covariate shift one could introduce is to add a new set of features that are not present in the training data with some probability at test time. Since the model has not seen these new features during training, we may assume that individual members of the ensemble will make random predictions (but the prediction of different models will be correlated) if the data points only contain the new features (the same assumption for features that the models did not learn). One can also introduce a covariate shift on existing features but that would require imposing a non-uniform distribution over the existing features. We leave the exploration of these directions to future works.
>
> We hope these answers clarify your question, and we are open to further discussion. Thank you again for your constructive feedback.

---

> > ### Comment · Reviewer_AyvM · 2023-11-16
> >
> > I would like to thank the authors for the clarifications and will maintain my initial assessment of the paper.

---

### Author Response · Authors · 2023-11-16
**Overall response**

We thank all the reviewers for their valuable time. We are glad that the reviewers found the results original, interesting, and timely.

We have modified Figure 1 and made minor revisions, and added some related works to the paper which are highlighted with orange in the text.

We will respond to each reviewer’s comments individually.

---

### Meta-Review · Area_Chair_B35K · 2023-12-14

**Metareview:**

Great paper: presents a methodology to describe the landscape of models learned from a given data distribution, that also induces a typology of the samples.
Each model is viewed as a bag of features (before the linear classification); after projection (to get rid of redundancies among samples), the features are clustered.
The samples are examined depending on whether they excite the features in a cluster.
The descriptive analysis of the samples is proposed, the claims are supported by "interventions" on the classes.

**Justification For Why Not Higher Score:**

NA

**Justification For Why Not Lower Score:**

Very clear; deep; rebuttal very well pondered.

---

### Decision · Program_Chairs · 2024-01-16

Accept (oral)